# PaSAL: A Deep Learning Pipeline for Pulmonary Artery-Vein Segmentation and Anatomical Labeling in Thoracic CT

**Jasper Eppink**[*1,2]                                                  JASPER.EPPINK@GMAIL.COM
**Hoel Kervadec**[*2] (iD)                                              H.T.G.KERVADEC@UVA.NL
**Julian van Capelleveen**[1]                                 J.C.VANCAPELLEVEEN@AMSTERDAMUMC.NL
**Joost Verhoeff**[1] (iD)                                    JOOST.VERHOEFF@AMSTERDAMUMC.NL
**Suresh Senan**[1] (iD)                                           S.SENAN@AMSTERDAMUMC.NL
**Omar Bohoudi**[1] (iD)                                         O.BOHOUDI@AMSTERDAMUMC.NL

[1] *Amsterdam UMC, Department of Radiation Oncology, Amsterdam, The Netherlands*

[2] *University of Amsterdam, Informatics Institute, Amsterdam, The Netherlands*

**Editors:** Accepted for publication at MIDL 2026

## Abstract

We present PaSAL, a deep learning pipeline for pulmonary artery-vein segmentation and anatomical labeling in thoracic CT. PaSAL combines an nnU-Net-based binary vessel segmentation model with a graph-based anatomical labeling framework that assigns 19 clinically defined vascular classes. The pipeline integrates vessel enhancement, skeletonization, and topology-aware label propagation to produce anatomically coherent outputs.

PaSAL is trained on the HiPaS and PTL public datasets and evaluated on an external set of 63 clinical scans from Amsterdam UMC. On HiPaS, PaSAL achieves Dice scores of 89.5% (arteries) and 88.1% (veins). On PTL, voxel-level anatomical labeling accuracy reaches 90.1% for arteries and 82.7% for veins. Expert review confirms high anatomical plausibility and clinical utility, while showing weak correlation between standard quantitative metrics and perceived quality.

To our knowledge, PaSAL is the first method to jointly perform artery-vein segmentation and anatomical labeling in CT. The results demonstrate robust performance across diverse anatomical presentations, including pre- and post-radiotherapy scans, and establish PaSAL as a useful baseline tool for vascular analysis in medical imaging.

**Keywords:** Pulmonary artery-vein segmentation, Pulmonary anatomical labeling, Thoracic CT, Deep learning, Graph-based learning

## 1. Introduction

Pulmonary vascular analysis plays an important role in diagnosing and monitoring lung disease, assessing treatment responses, and understanding long-term changes in pulmonary function. Artery-vein (A/V) segmentation in thoracic CT enables detection of emboli, assessment of pulmonary hypertension, and can help evaluate treatment-related vascular changes, such as damage or remodeling after radiotherapy. Anatomical labeling of vascular branches further enables regional quantification and longitudinal analysis of vascular remodeling, which is particularly relevant for studying radiation-induced lung injury.

---

[*] Contributed equally

Despite recent progress in deep learning for medical imaging, automated A/V segmentation and anatomical labeling remain challenging. Arteries and veins exhibit similar intensities in CT, especially in non-contrast scans, and distal branches are thin, tortuous, and prone to discontinuities. Accurate A/V differentiation requires integrating long-range structural context, which conventional CNNs struggle to capture. Anatomical labeling adds an additional layer of complexity, as labels must respect global topology across the entire vascular tree. Progress in this field is further hindered by the absence of publicly available datasets that provide both A/V segmentation and anatomical labels, forcing researchers to combine heterogeneous datasets and limiting generalization across patient populations, disease states, and acquisition protocols.

Existing approaches typically address either A/V segmentation or anatomical labeling in isolation. As a result, they do not produce fully structured vascular trees and are difficult to apply in large-scale or longitudinal studies where anatomically consistent quantification is essential. Furthermore, current datasets lack distal vessel annotations, preventing existing methods from producing complete vascular trees.

In this work, we introduce **PaSAL** (Pulmonary artery-vein Segmentation and Anatomical Labeling), a unified deep learning pipeline that jointly performs high-resolution A/V segmentation and 19-class anatomical labeling from CT. PaSAL integrates nnU-Net segmentation with a graph-based labeling framework and introduces several new components. These include hierarchical extended vessel targets, topology-aware skeleton reconnection, robust graph extraction and reorientation, and peripheral label propagation. Together, these components enable anatomically coherent predictions across the entire vascular tree. The pipeline is trained on the HiPaS and PTL datasets and evaluated on an external cohort of 63 clinical scans, from patients who underwent lung radiotherapy.

**Our main contributions are:** (i) the first unified end-to-end pipeline that bridges the gap between raw thoracic CT and structured, 19-class labeled vascular trees to enable anatomically consistent longitudinal analysis; (ii) new data engineering components and target refinements designed to address the distal annotation gap, including hierarchical extended vessel targets and an MST-inspired skeleton reconnection strategy for robust tree construction; (iii) a comprehensive validation on a clinical cohort of 63 scans from lung cancer patients, demonstrating system robustness across diverse pathologies and significant post-radiotherapy remodeling; and (iv) an analysis revealing a decoupling between standard overlap metrics and expert-perceived clinical quality in high-performance regimes, challenging current evaluation paradigms for complex vascular structures.

An overview of the PaSAL prediction pipeline is shown in Figure 1. Details on data preprocessing, model architectures, and post-processing steps are provided in the Methods section, with additional implementation details in the Appendix.

## 2. Related Work

### 2.1. Pulmonary Artery-Vein Segmentation

Early work on pulmonary artery-vein (A/V) segmentation primarily relied on CTPA and classical image-processing pipelines combining vesselness filtering, anatomical heuristics, or semi-automatic propagation (Wittram, 2007; Mekada et al., 2006; Buelow et al., 2005; Payer et al., 2016). Learning-based variants such as AdaBoost voxel classifiers (Ochs et al.,

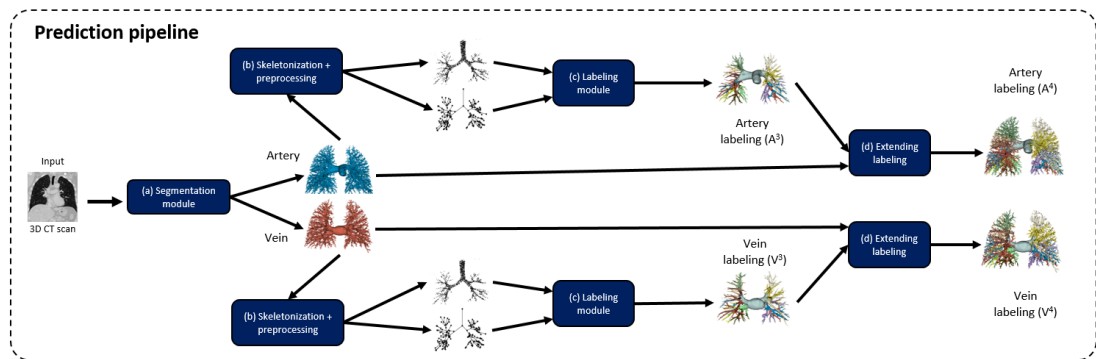

Figure 1: Overview of the PaSAL prediction pipeline. Each dark blue block represents one of the four main modules: (a) hierarchical segmentation, (b) preprocessing for anatomical labeling, (c) IPGN-based anatomical labeling, and (d) label propagation to peripheral vessels.

2007) and early graph-based methods (Charbonnier et al., 2015) improved robustness but remained limited by handcrafted features.

Deep learning substantially advanced pulmonary vessel segmentation. CNN-based approaches (Nardelli et al., 2018; Tetteh et al., 2020) improved voxel-level accuracy but often struggled with distal branches and global A/V distinction. Subsequent work emphasized better structural context via tubule-sensitive designs (Qin et al., 2021), geometry-aware or multi-resolution architectures (Pu et al., 2023; Pan et al., 2023), and hierarchical prediction schemes such as HiPaS (Chu et al., 2025). Transformers have also been explored for vessel modeling and A/V separation by leveraging long-range attention (Wu et al., 2023b,a; Lv et al., 2024).

## 2.2. Pulmonary Anatomical Vessel Labeling

Anatomical labeling assigns each vessel branch to a semantic pulmonary region and has been studied more extensively for airways than for vasculature. Prior work includes structure-aware CNN-GNN hybrids (Tan et al., 2021), topology-guided GNNs (Xie et al., 2024), and hypergraph-based tree models (Yu et al., 2022). These approaches capture hierarchical structure but generally rely on voxel-based predictions or predefined templates, limiting applicability to complex vascular trees.

Point-based and graph-based learning provide more flexible representations. Earlier work explored geometric matching of bronchial trees (Bülow et al., 2006), while recent point-cloud pipelines such as RibSeg demonstrated strong anatomical labeling performance for bony structures (Jin et al., 2023). Graph-based models further encode topological relationships but still require mechanisms for dense reconstruction.

Most relevant to our work is the Implicit Point-Graph Network (IPGN) (Xie et al., 2025), the first anatomical labeling framework developed specifically for pulmonary arteries and veins. IPGN integrates point-based and graph-based reasoning with an implicit field decoder to obtain dense, high-resolution anatomical labels for the full vascular tree.

## 3. Methods

### 3.1. Datasets

PaSAL is trained and evaluated on three datasets whose structural characteristics directly motivate several components of our pipeline. A key limitation across all training datasets is the absence of the smallest peripheral vessels in the provided artery/vein annotations. Addressing this gap is a central contribution of our pipeline, motivating the early presentation of the datasets.

For artery/vein segmentation we use the HiPaS dataset (Chu et al., 2025), which provides 250 chest CT volumes with artery and vein masks that cover vessels approximately up to mid-level branches. For anatomical labeling we use the Pulmonary Tree Labeling (PTL) dataset (Xie et al., 2025), containing 799 vascular and airway trees labeled with 19 anatomical classes, again limited to central and mid-peripheral vessels. Clinical generalization is assessed on an Amsterdam UMC cohort of 63 CT scans from 12 lung cancer patients imaged pre- and post-radiotherapy; predictions were scored by a radiation oncologist.

Table 1: Summary of datasets. "Annotation extent" describes the availability and depth of vessel labels; neither public dataset includes the smallest peripheral branches. [*]The 63 scans in the clinical evaluation dataset come from 12 distinct patients.

| Task | Source | Count | Usage | Annotation extent |
|------|--------|-------|-------|-------------------|
| A/V segmentation | HiPaS | 250 | Training/validation | Central–mid vessels; no distal branches |
| Anatomical labeling | PTL | 799 | Training/validation | Central–mid vessels; no distal branches |
| Clinical evaluation | AUMC | 63[*] | Clinical testing | No vessel annotations available |

All datasets contain scans of patients with pulmonary pathologies. The training datasets (HiPaS and PTL) include patients with diverse pathologies such as emboli and lung tumors. The clinical evaluation dataset (Amsterdam UMC) contains only lung cancer patients and includes scans acquired both before and after receiving radiotherapy. Because the public training datasets were provided in NIfTI/NPZ formats, DICOM metadata, including slice thickness and contrast phase, was unavailable. However, it is known from the original publications that the HiPaS dataset contains scans acquired with both contrast-enhanced (CECT) and non-contrast (NCCT) protocols. This prevents protocol-stratified training and analysis of whether performance disparities exist between these protocols, which is a key consideration given that contrast enhancement varies significantly between NCCT and CECT. For the in-house Amsterdam UMC clinical evaluation cohort, protocol information was available from DICOM headers. Additional dataset characteristics are summarized in Appendix A.

### 3.2. PaSAL pipeline overview

PaSAL produces anatomically labeled trees through four integrated stages. Unlike standard out-of-the-box implementations, we introduce non-trivial integration strategies: (i) hierarchical segmentation using extended Level-4 targets to capture peripheral vessels; (ii) a deterministic graph-extraction pipeline involving MST-inspired skeleton reconnection; (iii)

orientation-standardized IPGN labeling; and (iv) a watershed-based label propagation module to bridge the gap between central labels and distal anatomy. A schematic overview of the full prediction pipeline is provided in Figure 1 in the Introduction.

### 3.3. Hierarchical artery–vein segmentation

We adopt the hierarchical Salience-Transmission Segmentation (STS) framework of Chu et al. (2025), which predicts artery and vein masks across four vessel levels

$$[A^1, V^1], [A^2, V^2], [A^3, V^3], [A^4, V^4],$$

ranging from central to distal branches. HiPaS only provides a single vessel mask corresponding to Level 3, so we reconstruct the missing proximal (Levels 1–2) and distal (Level 4) targets using the provided vessel mask.

**Extended distal supervision (Level 4).** Level 3 corresponds to the original HiPaS annotations, which omit the smallest peripheral vessels. To expose the network to distal vessel morphology and reduce under-segmentation near the vascular periphery, we construct extended Level 4 targets by merging the HiPaS masks with TotalSegmentator vessel predictions and refining distal branches via constrained region growing seeded inside the predicted vessels (Fig. 2).

These extended targets are used exclusively during training to provide distal supervision and are excluded from all quantitative evaluation. They were reviewed by a clinical expert and deemed to be of sufficient quality, although minor localized inconsistencies may be present due to their partially automatic construction. As these inconsistencies are not systematic across the dataset, their influence during training is expected to be limited. All reported segmentation metrics are computed solely on the original HiPaS annotations, and Level 4 predictions are considered only in qualitative clinical assessment, where no distal ground truth is available.

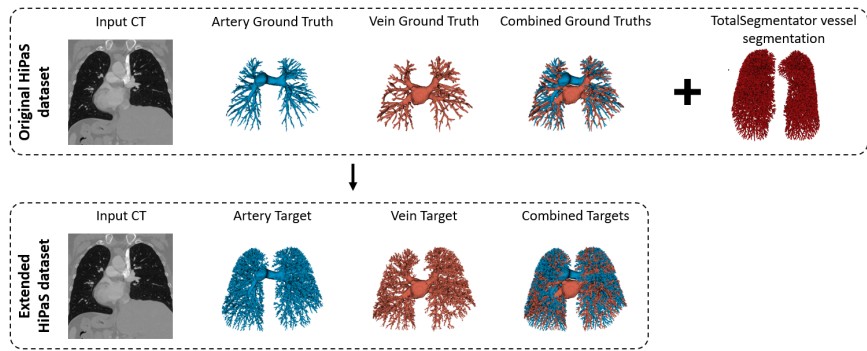

Figure 2: Construction of extended Level 4 targets on HiPaS by fusing the original labels with TotalSegmentator predictions and refining distal branches via region growing.

**Skeleton-based hierarchy construction (Levels 1-2).** To recover the missing proximal levels required by STS, we derive Levels 1 and 2 directly from the topology of the Level 3 vessel tree. We first extract vessel skeletons from the HiPaS masks using 3D medial-axis thinning (Lee et al., 1994), since skeletons explicitly encode branch order, path length, and bifurcations and are therefore a natural basis for hierarchical targets. However, raw skeletons often contain multiple disconnected components due to gaps and annotation inconsistencies. We address this by introducing a two-phase MST-inspired reconnection strategy (see Figure 3). First, in the mask-validated phase, nearest-neighbor nodes are linked only if the connecting edge lies predominantly ($\geq 90\%$) within the vessel mask. Second, a fallback phase ensures a single globally connected tree by adding shortest edges regardless of mask constraints. This deterministic process allows us to derive branch-order labels based on the distance from the largest average radius edge, which acts as the tree root.

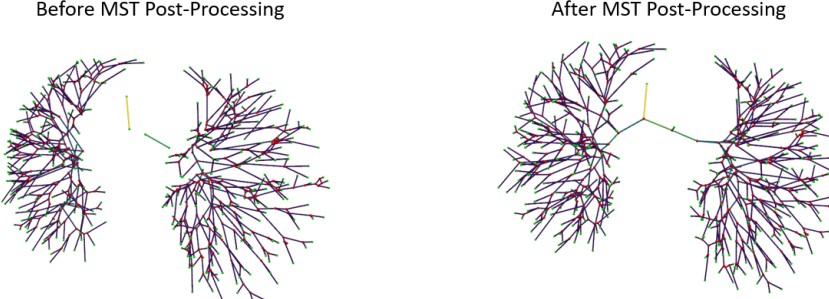

Figure 3: MST-style skeleton reconnection merges disconnected components into a single connected vascular tree.

**Hierarchical target assignment.** From the connected skeleton, voxels are assigned to Levels 1-3 using simple radius- and distance-based rules, producing the coarse-to-fine vessel priors required by STS. Combined with the extended Level 4 supervision, this yields the complete hierarchy used during training (Figure 4).

At each level $i \geq 2$, the model receives the CT volume, a Frangi vesselness map (Frangi et al., 1998), and predictions from the preceding level; Level 1 uses only CT and the Frangi vesselness map. We employ the standard 3D full-resolution nnU-Net (Isensee et al., 2021), training separate artery and vein models for each level.

### 3.4. Anatomical labeling

Anatomical labeling is performed by transforming the Level-3 vessel segmentations into connected trees suitable for graph-based learning. Vessel centerlines are extracted using 3D medial-axis thinning, followed by the MST-inspired reconnection strategy described in Section 3.3 to enforce global connectivity. The resulting graphs are deterministically reoriented by standardizing axis permutations and rotations to match the PTL coordinate convention.

We use the Implicit Point-Graph Network (IPGN) Xie et al. (2025) as the anatomical labeling backbone. IPGN combines point-based features with graph topology to predict

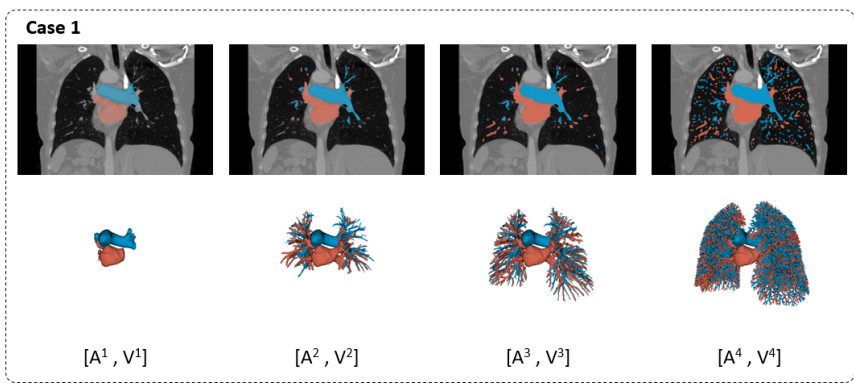

Figure 4: Four-level artery–vein target hierarchy for one HiPaS case. Levels 1–3 are evaluated; Level 4 provides distal supervision during training only.

19 anatomical classes. We employ the publicly released pre-trained artery and vein models without architectural modification to produce graph-, point-, and voxel-level labels; as IPGN itself is not a contribution of this work, we do not revisit its internal design. Because the PTL training data is limited to Level-3 vessels, labeling is consistently restricted to Level-3 segmentations across all datasets to ensure prediction reliability. A schematic overview of the IPGN framework is provided in Appendix B (Figure 6).

### 3.5. Label propagation to peripheral vessels

Because the IPGN training dataset is limited to targets corresponding to level 3 targets and therefore do not contain the distal peripheral vessels, we introduce a marker-based watershed algorithm to propagate these labels to the Level-4 peripheral segmentations. This step is essential for producing complete vascular trees, bridging the gap in existing datasets where ground truth for distal branches is unavailable. IPGN voxel predictions act as seed markers and the watershed operates within the union of Level 4 artery and vein masks (Van der Walt et al., 2014). This post hoc propagation step enables PaSAL to produce fully labeled vascular trees despite the lack of ground truth for peripheral branches. Propagated labels are used only for qualitative assessment, since no ground truth exists for these regions. Implementation details are given in Appendix B.

### 3.6. Training and evaluation

All segmentation models follow the default 3D full-resolution nnU-Net training pipeline (Isensee et al., 2021) with a combined Dice and cross-entropy loss and standard 3D augmentations. Artery and vein models are trained separately for each hierarchical level.

For anatomical labeling, we use the publicly released IPGN artery and vein models (Xie et al., 2025) without modification, but integrate them into our CT-based pipeline via the custom preprocessing and graph-construction steps described above.

Segmentation performance on HiPaS is evaluated using Dice, HD95, Sensitivity, and Precision, while anatomical labeling accuracy on PTL is assessed using voxel-, node-, and edge-level micro-averaged Dice following (Xie et al., 2025).

Beyond quantitative metrics, we assess clinical usefulness on the in-house radiotherapy cohort using a structured 0–5 scoring protocol completed by a radiation oncologist. For segmentation, scores reflect overall accuracy/robustness, peripheral branch completeness, and diagnostic usefulness; for labeling, they assess label consistency, proximal-distal correctness, and clinical interpretability (full criteria in Appendix D). We report mean expert scores and compute Spearman correlations between quantitative metrics and expert ratings to study how well overlap-based measures reflect perceived clinical utility. Given that the primary goal of the clinical evaluation was to establish clinical utility and practical applicability rather than protocol-specific performance analysis, differences in performance between NCCT and CECT protocols were not explicitly analyzed in the clinical cohort.

## 4. Results

We evaluate PaSAL on three tasks: (i) artery-vein segmentation on HiPaS, (ii) anatomical labeling on PTL, and (iii) full-pipeline clinical viability on a longitudinal radiotherapy cohort. All per-case metrics, extended visualizations, and correlation plots are provided in Appendix E.

### 4.1. Segmentation

Segmentation performance on the HiPaS test set (13 scans) is summarized in Table 2. Arteries consistently outperform veins across Dice, Sensitivity, Precision, and HD95, with Dice values around 90% for both structures. Slightly lower venous performance is attributable to weaker venous skeleton priors and less reliable graph-root selection during hierarchy construction. Several lower-Dice outliers correspond to anatomically valid distal branches missing from the HiPaS ground truth rather than true model errors.

Overall values appear lower than those reported by Chu et al. (2025); however, a direct comparison is not meaningful because the full target annotations used in their study were not released publicly. While the original scans are available, the publicly released annotations exclude distal vessels, resulting in a different target definition. In this work, segmentation metrics are therefore reported on the released Level-3 targets, which enable quantitative evaluation but do not capture performance on the full peripheral vascular tree. Consequently, we do not include a separate segmentation baseline, as comparisons restricted to these targets would neither reflect our primary objective of assessing the clinical usefulness of complete vascular tree predictions nor support our secondary objective of analyzing the relationship between quantitative metrics and expert-perceived clinical quality.

Expert evaluation of PaSAL's segmentation on nine HiPaS scans yielded high scores across all categories (Table 3). Ratings ranged from 3.6–4.0, indicating strong perceived accuracy, robustness, and practical clinical utility. Within this high-performance regime, voxelwise metrics did not correlate with expert judgment (maximum Spearman correlation $|\rho| = 0.52$, vein Dice vs. vessel branch abundance, $p = 0.15$; most $|\rho| < 0.25$, see Appendix Table 9 and Figs. 7–8). This suggests that for segmentation methods already achieving

Table 2: Segmentation performance on the HiPaS test set (13 scans). Values are mean (standard deviation).

| Metric | Artery | Vein |
|---|---|---|
| Dice | 90.0 (2.2) | 88.7 (2.5) |
| Sensitivity | 91.9 (3.4) | 90.0 (3.3) |
| Precision | 88.4 (4.4) | 87.8 (5.9) |
| HD95 (mm) | 3.62 (3.42) | 6.77 (6.46) |

Table 3: Expert assessment of artery/vein segmentation on nine HiPaS scans. Scores are mean (standard deviation) on a 0–5 scale.

| Category | Artery | Vein |
|---|---|---|
| Segmentation Accuracy and Robustness | 3.7 (0.7) | 3.6 (0.5) |
| Vessel Branch Abundance | 4.0 (0.5) | 3.9 (0.6) |
| Diagnostic Assistance | 4.0 (0.5) | 4.0 (0.7) |
| Mean Score | 3.9 (0.4) | 3.8 (0.5) |

Dice values near 90%, further improvements in overlap or distance metrics do not reliably reflect perceived clinical quality.

We note that PaSAL predicts vessels up to Level-3, and both quantitative metrics and expert assessments are therefore derived from these predictions. Certain expert criteria, including vessel branch abundance, also consider distal branches that are not predicted by the model, which may partly explain the weak correlations observed.

## 4.2. Anatomical Labeling

Labeling performance on the PTL test set (160 scans) is summarized in Table 4. Arteries consistently achieve higher voxel-, node-, and edge-level Dice than veins, reflecting greater geometric ambiguity and inter-patient variability in venous anatomy. To ensure a rigorous comparison, we re-evaluated the baseline implementation of Xie et al. (2025) using the exact same dataset splits and evaluation protocol. As shown in Table 4, PaSAL achieves labeling performance that is closely comparable to the baseline across all metrics for both arteries and veins, with only minor differences that are inconsistent in direction. Since identical IPGN model weights were used and no retraining was performed, these results indicate that enforcing graph connectivity does not degrade labeling accuracy, while providing a structured representation that may be useful for downstream and future vascular analyses.

Expert labeling assessments on 21 PTL scans (Table 5) yielded mean scores of 3.1–3.7, with the highest ratings for clinical interpretability, indicating that the labeled vascular trees are usable in clinical application. Correlations between Dice-based metrics and expert scores were consistently weak (maximum $|\rho| = 0.38$, vein voxel Dice vs. label consistency, $p > 0.05$; Appendix Table 12 and Figs. 10–11). Taken together, these findings show that voxelwise and graph-based Dice capture only a limited portion of what experts consider clinically important. In this high-performance regime, aiming for higher metric scores alone

Table 4: Anatomical labeling on the PTL test set (160 scans). Values are mean (std) Dice for PaSAL and the re-evaluated baseline using identical splits and metrics.

| Metric | Artery | Vein | Xie (A) | Xie (V) |
|---|---|---|---|---|
| Voxel Dice | 89.6 (3.8) | 83.1 (3.2) | 89.4 (4.0) | 83.0 (3.1) |
| Node Dice | 98.1 (2.1) | 94.9 (2.9) | 98.3 (2.1) | 95.3 (2.6) |
| Edge Dice | 90.7 (5.6) | 78.9 (4.8) | 90.4 (5.9) | 79.0 (4.5) |

Table 5: Expert assessment of anatomical labeling on 21 PTL scans. Scores are mean (standard deviation) on a 0–5 scale.

| Category | Artery | Vein |
|---|---|---|
| Label Consistency Across Branches | 3.1 (0.8) | 3.1 (0.8) |
| Correctness of Proximal vs. Distal Labeling | 3.3 (0.6) | 3.2 (0.7) |
| Usefulness for Clinical Interpretation | 3.7 (0.5) | 3.6 (0.5) |
| Mean Score | 3.4 (0.5) | 3.3 (0.5) |

is unlikely to translate into improved clinical utility or better expert-perceived quality, underscoring the need for evaluation criteria beyond conventional accuracy metrics.

### 4.3. Clinical Viability

We further evaluated full-pipeline performance, including label propagation to peripheral branches, on 63 longitudinal CT scans from 12 radiotherapy patients. Outputs remained anatomically coherent across timepoints, although substantial post-treatment deformation occasionally reduced local smoothness in distal branches. A representative baseline–follow-up pair is shown in Fig. 5.

A clinical expert assigned average scores of 3.4–3.9 across anatomical completeness, labeling plausibility, and practical clinical utility (Table 6). No scan scored below 3, and lower ratings were primarily associated with coarse voxel spacing or treatment-induced anatomical shifts rather than systematic limitations of the pipeline. Per-scan ratings are provided in Appendix E.

## 5. Discussion

This work introduced PaSAL, a fully automated pipeline for pulmonary artery-vein segmentation and anatomical labeling in thoracic CT. By combining hierarchical nnU-Net segmentation with IPGN-based graph labeling, PaSAL produces structured vascular representations suitable for anatomical and longitudinal analyses.

### 5.1. Interpretation of Results

PaSAL achieves high performance at level $[A^3, V^3]$, with Dice, Sensitivity, and Precision consistently above 90% for arteries and slightly lower for veins. Arterial predictions benefit from stronger morphological priors and more stable centerline extraction, whereas peripheral

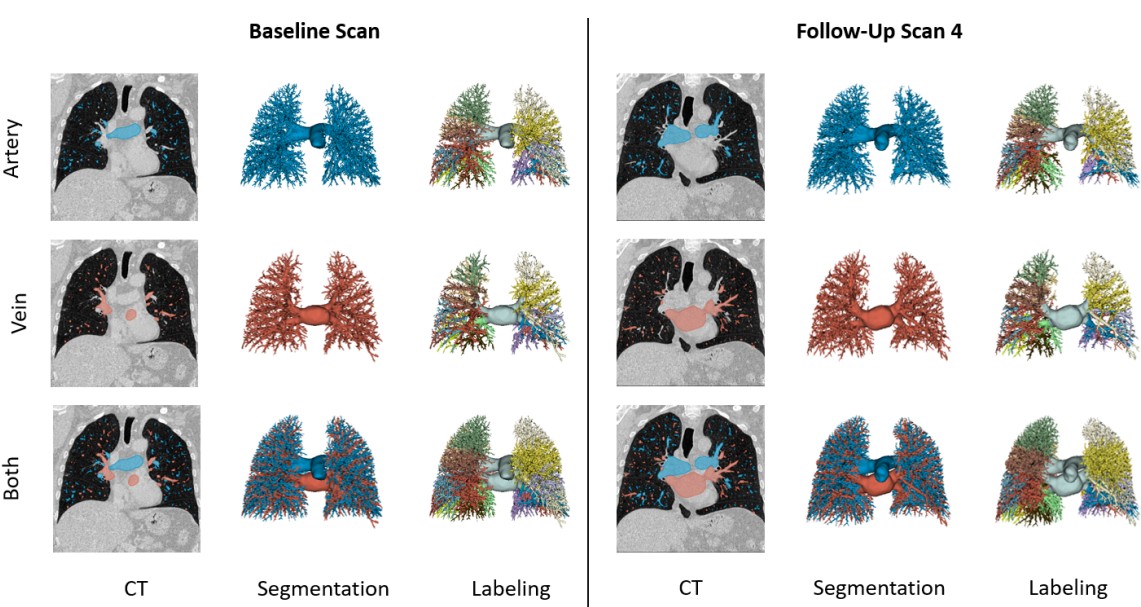

Figure 5: Pipeline predictions for a representative patient from the clinical cohort. Left: baseline scan; right: post-radiotherapy follow-up.

Table 6: Clinical expert evaluation on 63 longitudinal scans from 12 patients. Scores are mean (standard deviation) on a 0–5 scale.

| Category | Artery | Vein |
|---|---|---|
| Anatomical Completeness and Accuracy | 3.4 (0.7) | 3.5 (0.6) |
| Consistency and Plausibility of Labeling | 3.5 (0.6) | 3.5 (0.6) |
| Clinical Utility | 3.9 (0.3) | 3.9 (0.3) |
| Mean Score | 3.6 (0.5) | 3.6 (0.4) |

venous branches remain challenging. Expert ratings (3.4-3.9/5) confirm that the outputs are clinically meaningful across heterogeneous anatomies, including post-treatment scans, indicating that the pipeline is viable for downstream pulmonary research.

A central finding of this study is that no meaningful correlation was observed between expert-perceived quality and conventional overlap metrics once these metrics are already high within the modeled anatomical scope. This aligns with recent observations (Kofler et al., 2023) and suggests that marginal improvements in overlap or distance metrics do not necessarily translate into perceived clinical usefulness for central pulmonary vessels. As expert assessment also reflects aspects beyond what is captured by voxelwise metrics alone, expert-based or task-based evaluations remain essential for assessing real-world applicability.

## 5.2. Strengths and Novelty

PaSAL jointly performs artery-vein separation and anatomical labeling, a combination rarely addressed in prior work. To our knowledge, it is the first pipeline to integrate and extend multiple public datasets to produce coherent, labeled vascular trees suitable for downstream analysis. Its modular design enables independent updates to segmentation and labeling components, making PaSAL a flexible foundation for pulmonary vascular research.

## 5.3. Limitations

Several limitations merit discussion. (1) Arteries and veins were segmented with separate models, potentially introducing inconsistencies in ambiguous regions; a unified multiclass model may improve spatial coherence. (2) Training partially relied on automatically extended HiPaS labels, which introduce local imperfections, particularly in distal vessels. (3) The training data lacked acquisition metadata, preventing protocol-stratified analysis and limiting interpretability across heterogeneous scans. (4) The multi-stage pipeline (segmentation, skeletonization, graph extraction, labeling) is susceptible to error propagation and depends on heuristic thresholds. (5) The loss function does not explicitly prioritize distal vessels; topology-aware objectives (Chu et al., 2025) may improve performance on small-caliber branches. (6) Anatomical labeling was restricted to $[A^3, V^3]$, with deeper branches inferred heuristically, which may induce errors in densely branched regions.

## 5.4. Future Work

Future work includes integrating acquisition metadata for contrast-aware preprocessing and protocol-stratified evaluation; developing unified multi-task architectures that jointly predict masks, centerlines, and labels to reduce error propagation; and exploring topology-aware losses to better capture peripheral vessels. Extending PaSAL to incorporate oncological context (e.g., dose maps, tumor masks) may enhance robustness to altered anatomy and support longitudinal vascular studies.

## 6. Conclusion

PaSAL provides a fully automated, clinically viable approach for pulmonary artery-vein segmentation and anatomical labeling in thoracic CT. Evaluations on public datasets and a radiotherapy cohort show that the pipeline produces anatomically coherent vascular trees with strong quantitative performance and expert-confirmed usability, demonstrating that structured pulmonary vascular analysis is feasible on realistic clinical data.

A key conclusion of this work is that improvements in conventional segmentation metrics do not necessarily yield better expert-perceived or clinically relevant outputs once performance is already high within the anatomical scope covered by available ground truth. This highlights the need for evaluation frameworks that go beyond overlap-based metrics and emphasizes that pipelines intended for clinical or research use must be validated in terms of anatomical correctness and practical usability. Despite limitations such as incomplete metadata, label imperfections, and the multi-stage architecture, PaSAL provides a solid foundation for further development of unified and clinically oriented pulmonary vascular analysis methods.

## Acknowledgments

The authors thank the Amsterdam UMC Department of Radiation Oncology for providing clinical data and departmental support for this work.

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

## Appendix Overview

This appendix contains additional dataset characteristics (Appendix A), implementation details for anatomical labeling including the IPGN architecture schematic (Appendix B), segmentation training details (Appendix C), clinical evaluation criteria (Appendix D), and the complete set of quantitative and qualitative evaluation results (Appendix E). These materials are provided to support reproducibility and to complement the descriptions in Section 3.

## Appendix A. Dataset Details and Limitations

This appendix summarizes additional characteristics of the datasets used in PaSAL and their implications for generalizability.

**Image formats and matrix sizes**

The public HiPaS and PTL datasets were provided as `.npz` volumes (with additional `.graphml` files for PTL graphs), while the in-house Amsterdam UMC cohort was available as `.nii.gz` (NIfTI) files. In our pipeline, most scans have matrix sizes of $N \times 512 \times 512$, with a smaller number at $N \times 718 \times 718$, $N \times 768 \times 768$, or $N \times 1024 \times 1024$. These matrix sizes reflect the stored image dimensions used for model training and inference, and should not be interpreted as acquisition resolution.

**Missing acquisition metadata**

Because all datasets were provided as NIfTI/NPZ and not as DICOM, several acquisition parameters are unavailable:

- Slice thickness and slice-to-slice spacing (increment) are not encoded, which prevents stratified analyses of model performance as a function of through-plane resolution.

- The scanning protocol (e.g. non-contrast CT vs. CTPA) is not recorded in the files, so we cannot quantify performance differences between contrast-enhanced and non-contrast scans.

For HiPaS, a separate metadata file with voxel spacing values was available and was used for reporting spacing distributions, but thickness and increment remain unknown.

**Population, pathology, and class imbalance**

HiPaS consists primarily of Chinese patients and includes cases with a range of pulmonary pathologies. PTL provides derived vascular and airway trees without demographic metadata. The in-house Amsterdam UMC cohort contains lung cancer patients scanned before and after radiotherapy. Across all datasets:

- Vessel voxels form a very small fraction of the volume, leading to strong class imbalance between background and vessel classes.

- Pathology (e.g. pulmonary embolism, lung tumours, post-radiotherapy changes) is present to varying degrees.

These factors can bias quantitative metrics and should be kept in mind when interpreting results and assessing generalizability to new populations.

## Appendix B. Anatomical Labeling Implementation Details

This appendix provides additional implementation details for the anatomical labeling component based on the IPGN framework (Xie et al., 2025).

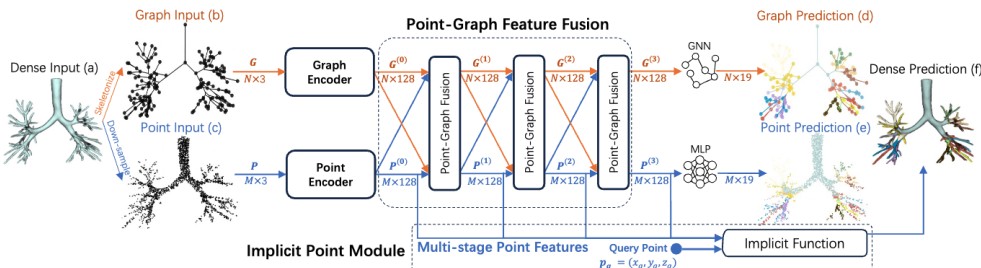

Figure 6: Schematic overview of the Implicit Point-Graph Network (IPGN) architecture for pulmonary tree labeling, created by Xie et al. (Xie et al., 2025). The network takes an extracted vessel segmentation (a), skeleton graph (b) and sampled point cloud (c) as input, encodes point and graph features, fuses them, and predicts anatomical labels at graph, point and voxel level.

### Graph extraction for labeling

Vessel skeletons are computed from the dense segmentation volumes using the thinning and MST-style reconnection procedures described in Section 3.3. Node coordinates are stored in physical space (voxel centers), and edges are defined between skeleton neighbours, resulting in a connected vessel graph for each artery and vein tree. These graphs are exported in .graphml format and paired with the corresponding binary segmentations, forming the inputs required by the pre-trained IPGN models.

### Label propagation to peripheral vessels

As discussed in the main text, the anatomical labeling model is limited to the extent of the original PTL segmentations (Level $[A^3, V^3]$). Our extended vessel masks (Level $[A^4, V^4]$) therefore contain unlabeled distal regions that are not directly covered by IPGN predictions.

To obtain fully labeled vascular trees for qualitative analysis, we applied a marker-based watershed algorithm over the Level 4 segmentations. IPGN voxel-level predictions at Level 3 served as seed markers, and labels were propagated to the remaining vessel voxels following the underlying distance transform. In practice, this produces anatomically plausible labels for most peripheral branches.

Because no ground truth labels are available for these distal regions, the propagated labels are used *only* for visualization and qualitative inspection. They are not used for supervised training and are excluded from all quantitative anatomical labeling metrics reported in the Results section.

## Appendix C. Segmentation Training Details

All nnU-Net models were trained in 3D full-resolution mode using the default nnU-Net training pipeline (Isensee et al., 2021). Key settings were:

- **Input configuration:** Multi-channel inputs as described in Section 3.3 (CT, Frangi vesselness, and previous-level artery/vein predictions where applicable).

- **Network and loss:** 3D full-resolution nnU-Net with the default combined Dice + cross-entropy loss.

- **Optimization:** Default nnU-Net optimizer and learning-rate schedule (stochastic gradient descent with Nesterov momentum and polynomial learning-rate decay).

- **Data augmentation:** nnU-Net's standard on-the-fly augmentations, including rotations, scaling, intensity transformations, Gaussian noise, blurring, and simulated low resolution.

- **Training regime:** Training performed separately for arteries and veins at each hierarchical level; only fold 0 was trained (instead of full 5-fold cross-validation) due to computational constraints, and the best checkpoint was selected based on validation loss.

No manual hyperparameter tuning beyond nnU-Net defaults was applied, apart from enabling multi-channel input for the salience-transmission setup.

## Appendix D. Clinical Evaluation Criteria

### Instructions for Clinical Expert - Segmentation

For each CT scan, please evaluate the segmentation output using the following three categories. Assign a score from 0 to 5 for each, where:

- 0 = Very poor
- 1 = Poor
- 2 = Fair
- 3 = Good
- 4 = Excellent
- 5 = Flawless

EVALUATION CRITERIA

**1. Segmentation Accuracy and Robustness**

- Are major vessels correctly segmented?
- Are there errors such as false splits or missing vessels?
- Is the segmentation reliable throughout the scan?

**2. Vessel Branch Abundance**

- Are sufficient peripheral branches captured?
- Is the segmentation too conservative or excessively noisy?
- Does it reflect the expected vascular complexity?

**3. Diagnostic Assistance**

- Could the segmentation help in diagnostic tasks (e.g., treatment planning)?
- Does it provide meaningful anatomical insight?
- Would it save time or effort in clinical workflows?

FORM TO FILL OUT

| Scan ID | Seg. accuracy & robustness | Branch abundance | Diagnostic assistance |
|---------|---------------------------|------------------|-----------------------|
| _____ | ____ / 5 | ____ / 5 | ____ / 5 |

**Instructions for Clinical Expert - Anatomical Labeling**

Same instructions and scoring scale as above.

EVALUATION CRITERIA

**1. Label Consistency Across Branches**

- Are connected vessel branches labeled coherently without unexpected label switches?

- Is anatomical continuity respected throughout bifurcations?

- Do labels remain stable across visually continuous regions?

**2. Correctness of Proximal vs. Distal Labeling**

- Are central (proximal) vessels labeled distinctly from peripheral (distal) ones?

- Does the labeling follow expected anatomical hierarchies (e.g., lobar, segmental vessels)?

- Are abrupt or incorrect zone transitions avoided?

**3. Usefulness for Clinical Interpretation**

- Does the labeling facilitate interpretation of anatomical regions?

- Could it assist in identifying perfusion territories, surgical zones, or radiation targets?

- Would this labeling support clinical tasks such as reporting, navigation, or planning?

FORM TO FILL OUT

| Scan ID | Label consistency | Proximal vs. distal | Clin. interpretability |
|---------|-------------------|---------------------|------------------------|
| _____ | ____ / 5 | ____ / 5 | ____ / 5 |

**Instructions for Clinical Expert - Full Pipeline**

Same instructions and scoring scale as above.

EVALUATION CRITERIA

1. **Anatomical Completeness and Accuracy**

   - Are both large and small vessel branches well represented and correctly segmented?

   - Do the anatomical labels match expected vascular structures and hierarchies?

   - Is there good alignment between the segmentation and the anatomical labeling?

2. **Consistency and Plausibility of Labeling**

   - Are labeled vessels anatomically consistent throughout the scan (e.g., no abrupt label changes)?

   - Is labeling coherent across bifurcations and throughout vascular trees?

   - Do labels correspond to known anatomical territories (e.g., lobes, segments)?

3. **Clinical Utility**

   - Could this combined output assist in clinical tasks such as treatment planning, surgical preparation, or diagnostic interpretation?

   - Does the visual output support clinical reasoning and decision-making?

   - Would it save time, reduce effort, or add value in a clinical workflow?

FORM TO FILL OUT

| Scan ID | Anat. completeness | Label plausibility | Clinical utility |
|---------|--------------------|--------------------|------------------|
| _____ | ____ / 5 | ____ / 5 | ____ / 5 |

Table 7: Per-subject quantitative segmentation metrics on the HiPaS test set.

| Subject | Dice (Artery) | HD95 (Artery) | Hausdorff (Artery) | Jaccard (Artery) | Precision (Artery) | Sensitivity (Artery) | Dice (Vein) | HD95 (Vein) | Hausdorff (Vein) | Jaccard (Vein) | Precision (Vein) | Sensitivity (Vein) |
|---|---|---|---|---|---|---|---|---|---|---|---|---|
| 007 | 0.92 | 1.41 | 31.59 | 0.85 | 0.89 | 0.95 | 0.89 | 5.83 | 33.00 | 0.81 | 0.85 | 0.94 |
| 027 | 0.90 | 1.41 | 22.38 | 0.82 | 0.89 | 0.91 | 0.90 | 3.32 | 35.13 | 0.82 | 0.90 | 0.91 |
| 029 | 0.89 | 1.00 | 22.00 | 0.80 | 0.83 | 0.96 | 0.91 | 2.24 | 46.05 | 0.84 | 0.89 | 0.94 |
| 036 | 0.88 | 1.41 | 35.74 | 0.79 | 0.82 | 0.95 | 0.90 | 25.08 | 76.07 | 0.82 | 0.89 | 0.91 |
| 058 | 0.94 | 1.00 | 95.68 | 0.88 | 0.94 | 0.93 | 0.91 | 1.41 | 84.08 | 0.83 | 0.93 | 0.89 |
| 063 | 0.92 | 1.00 | 36.19 | 0.86 | 0.91 | 0.93 | 0.89 | 3.61 | 78.55 | 0.81 | 0.88 | 0.90 |
| 071 | 0.93 | 1.41 | 104.93 | 0.87 | 0.94 | 0.92 | 0.87 | 8.19 | 110.66 | 0.77 | 0.82 | 0.93 |
| 164 | 0.89 | 6.00 | 288.38 | 0.81 | 0.91 | 0.87 | 0.89 | 1.41 | 25.63 | 0.81 | 0.94 | 0.85 |
| 174 | 0.89 | 8.49 | 45.22 | 0.81 | 0.89 | 0.90 | 0.90 | 6.40 | 192.15 | 0.82 | 0.93 | 0.87 |
| 189 | 0.88 | 11.18 | 221.41 | 0.79 | 0.88 | 0.89 | 0.86 | 11.22 | 41.00 | 0.75 | 0.83 | 0.88 |
| 190 | 0.87 | 4.12 | 74.05 | 0.77 | 0.81 | 0.94 | 0.82 | 5.66 | 95.12 | 0.70 | 0.73 | 0.94 |
| 247 | 0.88 | 5.00 | 31.83 | 0.79 | 0.90 | 0.86 | 0.88 | 6.93 | 30.48 | 0.79 | 0.93 | 0.84 |

Table 8: Per-subject qualitative segmentation scores on the HiPaS test set.

| Scan | Diagnostic Assistance (Artery) | Mean Score (Artery) | Segmentation Accuracy and Robustness (Artery) | Vessel Branch Abundance (Artery) | Diagnostic Assistance (Vein) | Mean Score (Vein) | Segmentation Accuracy and Robustness (Vein) | Vessel Branch Abundance (Vein) |
|---|---|---|---|---|---|---|---|---|
| 007 | 4 | 4.33 | 5 | 4 | 5 | 4.33 | 4 | 4 |
| 027 | 4 | 3.67 | 3 | 4 | 4 | 3.67 | 3 | 4 |
| 029 | 5 | 4.67 | 4 | 5 | 5 | 4.67 | 4 | 5 |
| 036 | 4 | 3.67 | 3 | 4 | 4 | 3.33 | 3 | 3 |
| 058 | 4 | 3.67 | 3 | 4 | 3 | 3.33 | 3 | 4 |
| 063 | 3 | 3.33 | 4 | 3 | 3 | 3.33 | 4 | 3 |
| 071 | 4 | 3.67 | 3 | 4 | 4 | 3.67 | 3 | 4 |
| 164 | 4 | 4.00 | 4 | 4 | 4 | 4.00 | 4 | 4 |
| 174 | 4 | 4.00 | 4 | 4 | 4 | 4.00 | 4 | 4 |

# Appendix E. Detailed Evaluation Results

This appendix provides the complete set of raw evaluation results corresponding to the experiments presented in Section 4. The tables are organized in the same order as the main Results section, and include both quantitative and qualitative assessments. These data are included to facilitate full transparency and reproducibility, and to allow other researchers to recreate or further analyze the presented plots.

## E.1. Segmentation Results

### E.1.1. Quantitative Evaluation (HiPaS Test Set)

The per-subject quantitative segmentation metrics for the HiPaS test set are reported in Table 7.

### E.1.2. Qualitative Evaluation (HiPaS Test Set)

The per-subject qualitative expert ratings for the HiPaS test set are summarized in Table 8.

### E.1.3. Comparison of Quantitative and Qualitative Evaluation

The correlation between quantitative segmentation metrics and qualitative expert scores on the HiPaS test set is given in Table 9.

The same relationships are visualized in Fig. 7 and Fig. 8.

## E.2. Anatomical Labeling Results

### E.2.1. Quantitative Evaluation on the PTL Test Set

Per-subject anatomical labeling accuracy on the PTL test set at the edge, node, and voxel levels is reported in Table 10.

Table 9: Correlation between quantitative segmentation metrics and qualitative expert scores on the HiPaS test set.

| Metric | Structure | ExpertCategory | Correlation | P-value |
|--------|-----------|----------------|-------------|---------|
| Dice | Artery | Segmentation Accuracy and Robustness | -0.17 | 6.55e-01 |
| Dice | Artery | Vessel Branch Abundance | -0.46 | 2.17e-01 |
| Dice | Artery | Diagnostic Assistance | -0.46 | 2.17e-01 |
| Dice | Artery | Mean Score | -0.45 | 2.19e-01 |
| Dice | Vein | Segmentation Accuracy and Robustness | -0.09 | 8.25e-01 |
| Dice | Vein | Vessel Branch Abundance | 0.52 | 1.53e-01 |
| Dice | Vein | Diagnostic Assistance | 0.19 | 6.18e-01 |
| Dice | Vein | Mean Score | 0.21 | 5.81e-01 |
| Sensitivity | Artery | Segmentation Accuracy and Robustness | 0.14 | 7.25e-01 |
| Sensitivity | Artery | Vessel Branch Abundance | 0.37 | 3.34e-01 |
| Sensitivity | Artery | Diagnostic Assistance | 0.37 | 3.34e-01 |
| Sensitivity | Artery | Mean Score | 0.25 | 5.10e-01 |
| Sensitivity | Vein | Segmentation Accuracy and Robustness | -0.09 | 8.25e-01 |
| Sensitivity | Vein | Vessel Branch Abundance | 0.21 | 5.89e-01 |
| Sensitivity | Vein | Diagnostic Assistance | 0.65 | 6.04e-02 |
| Sensitivity | Vein | Mean Score | 0.32 | 4.07e-01 |
| Precision | Artery | Segmentation Accuracy and Robustness | -0.31 | 4.16e-01 |
| Precision | Artery | Vessel Branch Abundance | -0.37 | 3.34e-01 |
| Precision | Artery | Diagnostic Assistance | -0.37 | 3.34e-01 |
| Precision | Artery | Mean Score | -0.44 | 2.39e-01 |
| Precision | Vein | Segmentation Accuracy and Robustness | 0.09 | 8.25e-01 |
| Precision | Vein | Vessel Branch Abundance | 0.09 | 8.19e-01 |
| Precision | Vein | Diagnostic Assistance | -0.26 | 5.02e-01 |
| Precision | Vein | Mean Score | -0.02 | 9.65e-01 |
| HD95 | Artery | Segmentation Accuracy and Robustness | -0.14 | 7.18e-01 |
| HD95 | Artery | Vessel Branch Abundance | 0.00 | 1.00e+00 |
| HD95 | Artery | Diagnostic Assistance | 0.00 | 1.00e+00 |
| HD95 | Artery | Mean Score | -0.27 | 4.91e-01 |
| HD95 | Vein | Segmentation Accuracy and Robustness | 0.22 | 5.74e-01 |
| HD95 | Vein | Vessel Branch Abundance | 0.46 | 2.13e-01 |
| HD95 | Vein | Diagnostic Assistance | -0.16 | 6.77e-01 |
| HD95 | Vein | Mean Score | 0.17 | 6.67e-01 |

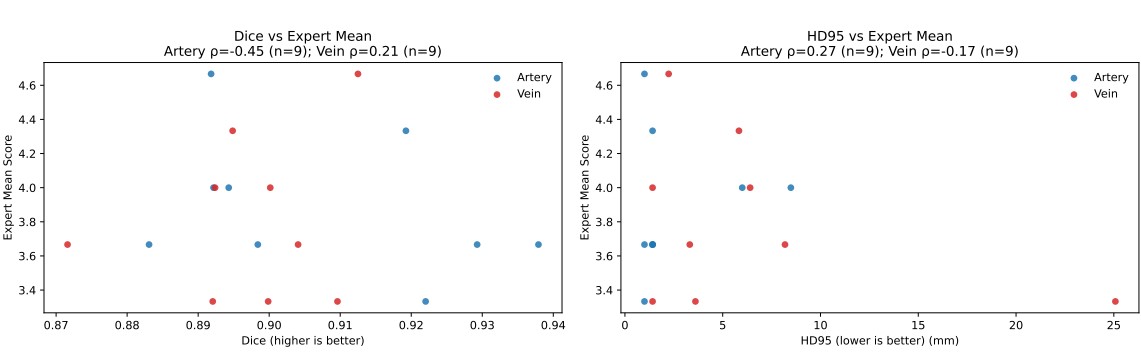

Figure 7: Spearman correlation between quantitative segmentation metrics and expert-assigned scores across the 9 overlapping HiPaS scans, shown separately for arteries (left) and veins (right).

Figure 8: Per-scan relationships between segmentation metrics and expert mean score for arteries and veins. Left: Dice vs. expert mean. Right: HD95 vs. expert mean (lower is better).

Table 10: Per-subject anatomical labeling accuracy on the PTL test set at edge, node, and voxel levels.

| Subject | Edge (Artery) | Node (Artery) | Voxel (Artery) | Edge (Vein) | Node (Vein) | Voxel (Vein) |
|---|---|---|---|---|---|---|
| 00007 | 0.95 | 0.98 | 0.89 | 0.81 | 0.94 | 0.83 |
| 00016 | 0.89 | 0.96 | 0.87 | 0.78 | 0.92 | 0.80 |
| 00019 | 0.87 | 0.97 | 0.86 | 0.80 | 0.95 | 0.83 |
| 00020 | 0.76 | 0.96 | 0.79 | 0.72 | 0.92 | 0.80 |
| 00022 | 0.94 | 0.98 | 0.90 | 0.84 | 0.94 | 0.83 |
| 00044 | 0.96 | 0.99 | 0.93 | 0.74 | 0.93 | 0.80 |

| Subject | Edge (Artery) | Node (Artery) | Voxel (Artery) | Edge (Vein) | Node (Vein) | Voxel (Vein) |
|---|---|---|---|---|---|---|
| 00054 | 0.95 | 0.99 | 0.92 | 0.83 | 0.97 | 0.86 |
| 00055 | 0.93 | 0.99 | 0.90 | 0.71 | 0.96 | 0.81 |
| 00078 | 0.88 | 0.98 | 0.89 | 0.86 | 0.98 | 0.85 |
| 00117 | 0.95 | 0.99 | 0.94 | 0.83 | 0.96 | 0.80 |
| 00125 | 0.85 | 0.97 | 0.86 | 0.80 | 0.94 | 0.83 |
| 00133 | 0.86 | 0.96 | 0.90 | 0.70 | 0.94 | 0.82 |
| 00138 | 0.93 | 0.99 | 0.90 | 0.79 | 0.95 | 0.84 |
| 00144 | 0.90 | 0.98 | 0.90 | 0.83 | 0.95 | 0.85 |
| 00153 | 0.96 | 0.99 | 0.93 | 0.85 | 0.95 | 0.85 |
| 00154 | 0.87 | 0.98 | 0.88 | 0.77 | 0.94 | 0.83 |
| 00162 | 0.86 | 0.99 | 0.87 | 0.85 | 0.99 | 0.86 |
| 00163 | 0.86 | 0.98 | 0.86 | 0.80 | 0.97 | 0.85 |
| 00164 | 0.92 | 0.99 | 0.90 | 0.77 | 0.95 | 0.80 |
| 00169 | 0.89 | 0.99 | 0.88 | 0.73 | 0.95 | 0.80 |
| 00172 | 0.91 | 1.00 | 0.93 | 0.81 | 0.98 | 0.84 |
| 00173 | 0.85 | 0.98 | 0.83 | 0.83 | 0.97 | 0.85 |
| 00176 | 0.91 | 0.99 | 0.91 | 0.80 | 0.98 | 0.86 |
| 00177 | 0.85 | 0.98 | 0.88 | 0.78 | 0.97 | 0.85 |
| 00179 | 0.85 | 0.97 | 0.88 | 0.83 | 0.98 | 0.87 |
| 00189 | 0.91 | 0.99 | 0.89 | 0.80 | 0.97 | 0.84 |
| 00192 | 0.97 | 1.00 | 0.93 | 0.77 | 0.97 | 0.86 |
| 00198 | 0.84 | 0.98 | 0.87 | 0.69 | 0.96 | 0.80 |
| 00206 | 0.93 | 0.98 | 0.91 | 0.78 | 0.92 | 0.77 |
| 00239 | 0.77 | 0.95 | 0.84 | 0.72 | 0.93 | 0.77 |
| 00247 | 0.90 | 0.97 | 0.91 | 0.72 | 0.85 | 0.79 |
| 00256 | 0.91 | 0.99 | 0.93 | 0.75 | 0.96 | 0.84 |
| 00258 | 0.94 | 1.00 | 0.90 | 0.82 | 0.98 | 0.84 |
| 00297 | 0.95 | 1.00 | 0.93 | 0.78 | 0.97 | 0.84 |
| 00361 | 0.95 | 1.00 | 0.93 | 0.76 | 0.95 | 0.85 |
| 00364 | 0.78 | 0.96 | 0.82 | 0.69 | 0.95 | 0.78 |
| 00368 | 0.85 | 0.98 | 0.86 | 0.75 | 0.95 | 0.81 |
| 00373 | 0.89 | 0.98 | 0.88 | 0.78 | 0.97 | 0.84 |
| 00382 | 0.90 | 0.99 | 0.91 | 0.82 | 0.98 | 0.85 |
| 00396 | 0.91 | 0.99 | 0.90 | 0.79 | 0.97 | 0.86 |
| 00407 | 0.91 | 0.99 | 0.90 | 0.79 | 0.96 | 0.85 |
| 00416 | 0.94 | 0.99 | 0.93 | 0.81 | 0.97 | 0.86 |
| 00431 | 0.96 | 1.00 | 0.93 | 0.83 | 0.98 | 0.88 |
| 00506 | 0.94 | 0.99 | 0.91 | 0.75 | 0.94 | 0.83 |
| 00560 | 0.94 | 0.99 | 0.93 | 0.83 | 0.96 | 0.87 |
| 00561 | 0.89 | 0.98 | 0.89 | 0.75 | 0.96 | 0.81 |
| 00570 | 0.91 | 0.98 | 0.91 | 0.80 | 0.96 | 0.86 |
| 00571 | 0.95 | 0.99 | 0.91 | 0.83 | 0.98 | 0.86 |
| 00575 | 0.95 | 1.00 | 0.92 | 0.75 | 0.96 | 0.82 |
| 00583 | 0.95 | 0.99 | 0.91 | 0.83 | 0.96 | 0.87 |
| 00589 | 0.94 | 1.00 | 0.93 | 0.74 | 0.97 | 0.82 |
| 00600 | 0.98 | 1.00 | 0.94 | 0.83 | 0.96 | 0.87 |
| 00702 | 0.86 | 0.99 | 0.88 | 0.84 | 0.98 | 0.87 |
| 00717 | 0.96 | 1.00 | 0.93 | 0.84 | 0.97 | 0.85 |
| 00729 | 0.93 | 0.99 | 0.91 | 0.79 | 0.96 | 0.83 |
| 00731 | 0.94 | 1.00 | 0.92 | 0.78 | 0.96 | 0.83 |
| 00732 | 0.95 | 0.99 | 0.93 | 0.84 | 0.96 | 0.86 |
| 00733 | 0.94 | 1.00 | 0.89 | 0.81 | 0.97 | 0.85 |
| 00742 | 0.86 | 0.98 | 0.90 | 0.75 | 0.96 | 0.82 |
| 00758 | 0.93 | 0.98 | 0.91 | 0.79 | 0.86 | 0.82 |
| 00773 | 0.96 | 0.99 | 0.92 | 0.83 | 0.92 | 0.83 |
| 00784 | 0.79 | 0.86 | 0.77 | 0.75 | 0.83 | 0.78 |
| 00796 | 0.82 | 0.95 | 0.82 | 0.76 | 0.94 | 0.79 |
| 00797 | 0.94 | 1.00 | 0.91 | 0.79 | 0.96 | 0.82 |
| 00799 | 0.97 | 0.98 | 0.94 | 0.84 | 0.89 | 0.83 |
| 00803 | 0.91 | 0.96 | 0.88 | NaN | NaN | NaN |

| Subject | Edge (Artery) | Node (Artery) | Voxel (Artery) | Edge (Vein) | Node (Vein) | Voxel (Vein) |
|---|---|---|---|---|---|---|
| 00804 | 0.94 | 0.98 | 0.91 | 0.80 | 0.87 | 0.82 |
| 00819 | 0.95 | 0.99 | 0.93 | 0.81 | 0.96 | 0.84 |
| 00821 | 0.90 | 0.97 | 0.89 | 0.73 | 0.92 | 0.78 |
| 00823 | 0.95 | 1.00 | 0.94 | 0.84 | 0.97 | 0.87 |
| 00830 | 0.94 | 0.99 | 0.93 | 0.82 | 0.97 | 0.86 |
| 00850 | 0.85 | 0.97 | 0.87 | 0.81 | 0.96 | 0.85 |
| 00854 | 0.92 | 0.99 | 0.91 | 0.82 | 0.96 | 0.87 |
| 00859 | 0.89 | 0.98 | 0.88 | 0.75 | 0.94 | 0.81 |
| 00863 | 0.93 | 0.99 | 0.93 | 0.72 | 0.93 | 0.77 |
| 00866 | 0.89 | 0.98 | 0.88 | 0.81 | 0.96 | 0.84 |
| 00874 | 0.95 | 0.99 | 0.94 | 0.79 | 0.95 | 0.85 |
| 00882 | 0.94 | 0.99 | 0.92 | 0.81 | 0.94 | 0.87 |
| 00884 | 0.91 | 0.97 | 0.90 | 0.82 | 0.95 | 0.85 |
| 00890 | 0.86 | 0.97 | 0.86 | 0.78 | 0.91 | 0.80 |
| 00891 | 0.91 | 0.97 | 0.89 | 0.73 | 0.92 | 0.82 |
| 00892 | 0.96 | 1.00 | 0.92 | 0.78 | 0.94 | 0.80 |
| 00907 | 0.91 | 0.98 | 0.90 | 0.75 | 0.94 | 0.81 |
| 00912 | 0.93 | 1.00 | 0.92 | 0.86 | 0.98 | 0.88 |
| 00935 | 0.93 | 0.99 | 0.91 | 0.81 | 0.95 | 0.85 |
| 00936 | 0.96 | 1.00 | 0.94 | 0.86 | 0.98 | 0.86 |
| 00943 | 0.90 | 0.98 | 0.89 | 0.71 | 0.94 | 0.79 |
| 00952 | 0.92 | 0.99 | 0.89 | 0.81 | 0.96 | 0.88 |
| 00957 | 0.92 | 0.98 | 0.92 | 0.79 | 0.95 | 0.84 |
| 00959 | 0.87 | 0.96 | 0.87 | 0.72 | 0.89 | 0.78 |
| 00982 | 0.69 | 0.89 | 0.77 | 0.62 | 0.86 | 0.67 |
| 00983 | 0.89 | 0.96 | 0.87 | 0.81 | 0.95 | 0.86 |
| 01114 | 0.93 | 1.00 | 0.89 | 0.81 | 0.97 | 0.83 |
| 01116 | 0.85 | 0.97 | 0.81 | 0.77 | 0.95 | 0.81 |
| 01131 | 0.92 | 0.99 | 0.89 | 0.77 | 0.97 | 0.82 |
| 01133 | 0.91 | 0.99 | 0.89 | 0.75 | 0.96 | 0.81 |
| 01138 | 0.95 | 1.00 | 0.92 | 0.80 | 0.97 | 0.86 |
| 01180 | 0.71 | 0.88 | 0.71 | 0.71 | 0.91 | 0.70 |
| 01191 | 0.89 | 0.99 | 0.89 | 0.82 | 0.97 | 0.86 |
| 01193 | 0.84 | 0.98 | 0.87 | 0.79 | 0.96 | 0.85 |
| 01207 | 0.87 | 0.98 | 0.85 | 0.79 | 0.96 | 0.84 |
| 01231 | 0.88 | 0.97 | 0.88 | 0.73 | 0.91 | 0.77 |
| 01258 | 0.91 | 0.99 | 0.89 | 0.76 | 0.96 | 0.82 |
| 01282 | 0.88 | 0.99 | 0.87 | 0.81 | 0.97 | 0.82 |
| 01324 | 0.89 | 0.95 | 0.84 | 0.79 | 0.91 | 0.78 |
| 01365 | 0.96 | 1.00 | 0.93 | 0.86 | 0.97 | 0.86 |
| 01373 | 0.96 | 0.99 | 0.94 | 0.85 | 0.94 | 0.88 |
| 01374 | 0.97 | 1.00 | 0.93 | 0.86 | 0.97 | 0.84 |
| 01381 | 0.95 | 0.98 | 0.92 | 0.80 | 0.88 | 0.85 |
| 01382 | 0.96 | 0.99 | 0.94 | 0.79 | 0.93 | 0.81 |
| 01390 | 0.95 | 0.98 | 0.92 | 0.79 | 0.92 | 0.81 |
| 01392 | 0.83 | 0.93 | 0.83 | 0.76 | 0.92 | 0.81 |
| 01395 | 0.95 | 0.99 | 0.94 | 0.82 | 0.97 | 0.84 |
| 01475 | 0.92 | 0.99 | 0.92 | 0.74 | 0.96 | 0.85 |
| 01486 | 0.91 | 0.98 | 0.90 | 0.82 | 0.97 | 0.85 |
| 01502 | 0.85 | 0.94 | 0.84 | 0.79 | 0.89 | 0.84 |
| 01505 | 0.96 | 1.00 | 0.93 | 0.86 | 0.98 | 0.87 |
| 01506 | 0.96 | 0.99 | 0.93 | 0.77 | 0.91 | 0.83 |
| 01515 | 0.97 | 1.00 | 0.92 | 0.84 | 0.98 | 0.85 |
| 01516 | 0.92 | 0.98 | 0.91 | 0.84 | 0.95 | 0.84 |
| 01518 | 0.92 | 0.99 | 0.91 | 0.81 | 0.97 | 0.88 |
| 01534 | 0.96 | 0.99 | 0.92 | 0.77 | 0.91 | 0.80 |
| 01535 | 0.92 | 0.99 | 0.89 | 0.78 | 0.97 | 0.84 |
| 01601 | 0.82 | 0.92 | 0.87 | 0.76 | 0.91 | 0.82 |
| 01602 | 0.93 | 0.99 | 0.88 | 0.86 | 0.98 | 0.85 |
| 01610 | 0.93 | 0.97 | 0.92 | 0.77 | 0.90 | 0.77 |

*Continued on next page*

| Subject | Edge (Artery) | Node (Artery) | Voxel (Artery) | Edge (Vein) | Node (Vein) | Voxel (Vein) |
|---|---|---|---|---|---|---|
| 01624 | 0.96 | 0.99 | 0.92 | 0.81 | 0.92 | 0.83 |
| 01627 | 0.87 | 0.98 | 0.89 | 0.67 | 0.94 | 0.81 |
| 01640 | 0.96 | 0.99 | 0.93 | 0.84 | 0.92 | 0.85 |
| 01659 | 0.97 | 1.00 | 0.93 | 0.82 | 0.95 | 0.86 |
| 01667 | 0.95 | 0.99 | 0.92 | 0.81 | 0.95 | 0.85 |
| 01679 | 0.96 | 0.99 | 0.92 | 0.86 | 0.95 | 0.86 |
| 01699 | 0.90 | 0.94 | 0.87 | 0.80 | 0.92 | 0.82 |
| 01706 | 0.88 | 0.98 | 0.88 | 0.68 | 0.95 | 0.80 |
| 01710 | 0.95 | 0.99 | 0.93 | 0.80 | 0.96 | 0.86 |
| 01711 | 0.97 | 1.00 | 0.93 | 0.80 | 0.88 | 0.84 |
| 01712 | 0.86 | 0.97 | 0.87 | 0.69 | 0.91 | 0.76 |
| 01714 | 0.90 | 0.96 | 0.89 | 0.85 | 0.94 | 0.85 |
| 01715 | 0.87 | 0.98 | 0.90 | 0.77 | 0.96 | 0.83 |
| 01718 | 0.96 | 0.99 | 0.95 | 0.77 | 0.92 | 0.78 |
| 01724 | 0.92 | 0.99 | 0.92 | 0.83 | 0.98 | 0.84 |
| 01747 | 0.96 | 0.99 | 0.90 | 0.81 | 0.94 | 0.84 |
| 01801 | 0.91 | 0.96 | 0.89 | 0.88 | 0.94 | 0.87 |
| 01812 | 0.97 | 1.00 | 0.93 | 0.79 | 0.98 | 0.83 |
| 01813 | 0.95 | 1.00 | 0.90 | 0.75 | 0.97 | 0.81 |
| 01818 | 0.90 | 0.98 | 0.91 | 0.80 | 0.97 | 0.83 |
| 01842 | 0.88 | 0.97 | 0.86 | 0.76 | 0.96 | 0.80 |
| 01858 | 0.89 | 0.99 | 0.89 | 0.76 | 0.98 | 0.83 |
| 01866 | 0.91 | 0.98 | 0.90 | 0.78 | 0.97 | 0.83 |
| 01867 | 0.87 | 0.99 | 0.92 | 0.79 | 0.97 | 0.86 |
| 01875 | 0.87 | 0.99 | 0.90 | 0.75 | 0.98 | 0.83 |
| 01879 | 0.95 | 0.99 | 0.93 | 0.83 | 0.98 | 0.85 |
| 01892 | 0.84 | 0.97 | 0.87 | 0.81 | 0.96 | 0.86 |
| 01893 | 0.88 | 0.98 | 0.89 | 0.68 | 0.96 | 0.78 |
| 01918 | 0.62 | 0.90 | 0.74 | 0.63 | 0.91 | 0.77 |
| 01925 | 0.91 | 0.99 | 0.88 | 0.80 | 0.96 | 0.87 |
| 01928 | 0.90 | 0.99 | 0.88 | 0.84 | 0.98 | 0.85 |
| 01948 | 0.95 | 1.00 | 0.92 | 0.82 | 0.97 | 0.84 |
| 01950 | 0.95 | 0.99 | 0.92 | 0.89 | 0.97 | 0.87 |
| 01970 | 0.87 | 0.99 | 0.86 | 0.79 | 0.98 | 0.85 |

Representative qualitative examples of high- and lower-performing cases are shown in Fig. 9.

### E.3. Qualitative Evaluation (PTL Test Set)

The per-scan qualitative expert ratings for the PTL test set are summarized in Table 11.

#### E.3.1. Comparison of Quantitative and Qualitative Evaluation

The correlation between anatomical labeling metrics and qualitative expert scores on the PTL test set is reported in Table 12.

Table 11: Per-scan qualitative expert ratings on the PTL test set.

| Scan | Correctness of Proximal vs. Distal Labeling (Artery) | Label Consistency Across Branches (Artery) | Mean Score (Artery) | Usefulness for Clinical Interpretation (Artery) | Correctness of Proximal vs. Distal Labeling (Vein) | Label Consistency Across Branches (Vein) | Mean Score (Vein) | Usefulness for Clinical Interpretation (Vein) |
|---|---|---|---|---|---|---|---|---|
| 00016 | 3 | 3 | 3.33 | 4 | 4 | 3 | 3.67 | 4 |
| 00054 | 3 | 2 | 2.67 | 3 | 4 | 4 | 4.00 | 4 |
| 00055 | 2 | 4 | 3.33 | 4 | 2 | 2 | 2.33 | 3 |
| 00078 | 4 | 4 | 4.00 | 4 | 3 | 3 | 3.33 | 4 |
| 00138 | 3 | 2 | 2.67 | 3 | 4 | 4 | 3.67 | 4 |
| 00176 | 3 | 4 | 3.67 | 4 | 4 | 3 | 3.67 | 3 |
| 00177 | 3 | 3 | 3.00 | 3 | 4 | 4 | 4.00 | 4 |
| 00192 | 4 | 2 | 3.33 | 4 | 3 | 3 | 3.33 | 4 |
| 00206 | 3 | 3 | 3.00 | 3 | 3 | 2 | 2.67 | 3 |
| 00297 | 4 | 3 | 3.67 | 4 | 2 | 4 | 3.33 | 4 |
| 00364 | 4 | 4 | 4.00 | 4 | 4 | 4 | 4.00 | 4 |
| 00396 | 2 | 2 | 2.33 | 3 | 3 | 2 | 2.67 | 3 |
| 00407 | 3 | 3 | 3.33 | 4 | 3 | 3 | 3.67 | 4 |
| 00560 | 3 | 4 | 3.67 | 4 | 3 | 3 | 3.00 | 3 |
| 00561 | 4 | 3 | 3.67 | 3 | 4 | 2 | 3.33 | 4 |
| 00575 | 4 | 4 | 4.00 | 4 | 3 | 3 | 3.00 | 3 |
| 00589 | 3 | 3 | 3.33 | 4 | 3 | 3 | 3.00 | 3 |
| 00731 | 3 | 2 | 2.67 | 3 | 4 | 3 | 3.67 | 4 |
| 00732 | 4 | 3 | 3.67 | 4 | 4 | 4 | 4.00 | 4 |
| 00733 | 4 | 4 | 4.00 | 4 | 2 | 2 | 2.33 | 3 |
| 00758 | 3 | 3 | 3.33 | 4 | 3 | 3 | 3.33 | 4 |

Table 12: Correlation between anatomical labeling metrics and qualitative expert scores on the PTL test set.

| Metric | Structure | ExpertCategory | Correlation | P-value |
|---|---|---|---:|---|
| Voxel | Artery | Label Consistency Across Branches | -0.26 | 2.62e-01 |
| Voxel | Artery | Correctness of Proximal vs. Distal Labeling | 0.03 | 9.03e-01 |
| Voxel | Artery | Usefulness for Clinical Interpretation | 0.18 | 4.26e-01 |
| Voxel | Artery | Mean Score | -0.06 | 7.83e-01 |
| Voxel | Vein | Label Consistency Across Branches | 0.38 | 9.20e-02 |
| Voxel | Vein | Correctness of Proximal vs. Distal Labeling | 0.09 | 6.86e-01 |
| Voxel | Vein | Usefulness for Clinical Interpretation | 0.02 | 9.44e-01 |
| Voxel | Vein | Mean Score | 0.24 | 2.96e-01 |
| Node | Artery | Label Consistency Across Branches | -0.16 | 4.96e-01 |
| Node | Artery | Correctness of Proximal vs. Distal Labeling | 0.18 | 4.26e-01 |
| Node | Artery | Usefulness for Clinical Interpretation | 0.15 | 5.16e-01 |
| Node | Artery | Mean Score | 0.04 | 8.48e-01 |
| Node | Vein | Label Consistency Across Branches | 0.08 | 7.43e-01 |
| Node | Vein | Correctness of Proximal vs. Distal Labeling | -0.09 | 6.86e-01 |
| Node | Vein | Usefulness for Clinical Interpretation | -0.15 | 5.28e-01 |
| Node | Vein | Mean Score | -0.08 | 7.17e-01 |
| Edge | Artery | Label Consistency Across Branches | -0.32 | 1.61e-01 |
| Edge | Artery | Correctness of Proximal vs. Distal Labeling | 0.18 | 4.24e-01 |
| Edge | Artery | Usefulness for Clinical Interpretation | 0.13 | 5.64e-01 |
| Edge | Artery | Mean Score | -0.05 | 8.28e-01 |
| Edge | Vein | Label Consistency Across Branches | 0.27 | 2.41e-01 |
| Edge | Vein | Correctness of Proximal vs. Distal Labeling | 0.02 | 9.17e-01 |
| Edge | Vein | Usefulness for Clinical Interpretation | 0.11 | 6.25e-01 |
| Edge | Vein | Mean Score | 0.19 | 4.03e-01 |

Table 13: Qualitative evaluation of clinical viability on the in-house longitudinal dataset.

| Scan | Anatomical Completeness and Accuracy (Artery) | Clinical Utility (Artery) | Consistency and Plausibility of Labeling (Artery) | Mean Score (Artery) | Anatomical Completeness and Accuracy (Vein) | Clinical Utility (Vein) | Consistency and Plausibility of Labeling (Vein) | Mean Score (Vein) |
|---|---|---|---|---|---|---|---|---|
| P54-BASELINE-SCAN | 4 | 4 | 4 | 4.00 | 4 | 4 | 3 | 3.67 |
| P54-FU1-3M-SCAN | 4 | 4 | 3 | 3.67 | 4 | 4 | 4 | 4.00 |
| P54-FU2-7M-SCAN | 2 | 4 | 3 | 3.00 | 2 | 4 | 3 | 3.00 |
| P56-Baseline-1.2m-SCAN | 3 | 4 | 4 | 3.67 | 3 | 4 | 4 | 3.67 |
| P56-FU1-3.0m-SCAN | 3 | 4 | 3 | 3.33 | 3 | 4 | 4 | 3.67 |
| P56-FU2-13.4m-SCAN | 2 | 4 | 2 | 2.67 | 3 | 4 | 2 | 3.00 |
| P56-FU3-22.7m-SCAN | 2 | 3 | 2 | 2.33 | 3 | 4 | 2 | 3.00 |
| P60-BASELINE-SCAN | 4 | 4 | 4 | 4.00 | 4 | 3 | 4 | 3.67 |
| P60-FU1-3.2-SCAN | 4 | 4 | 4 | 4.00 | 4 | 4 | 4 | 4.00 |
| P60-FU1-9.4-SCAN | 4 | 4 | 4 | 4.00 | 4 | 4 | 3 | 3.67 |
| P60-FU2-9.4-SCAN | 4 | 4 | 4 | 4.00 | 4 | 4 | 4 | 4.00 |
| P60-FU2-22.8-SCAN | 3 | 3 | 3 | 3.00 | 3 | 4 | 4 | 3.67 |
| P76 Baseline-1.3m-SCAN | 4 | 4 | 3 | 3.67 | 4 | 3 | 3 | 3.33 |
| P76-FU 27.2m-SCAN | 3 | 4 | 3 | 3.33 | 3 | 4 | 3 | 3.33 |
| P76-FU1-3.0m-SCAN | 4 | 4 | 3 | 3.67 | 4 | 4 | 3 | 3.67 |
| P76-FU2-18.1m-SCAN | 3 | 3 | 3 | 3.00 | 3 | 4 | 3 | 3.33 |
| P80-BASELINE-SCAN | 4 | 4 | 4 | 4.00 | 4 | 4 | 4 | 4.00 |
| P80-FU1-SCAN-11m | 3 | 4 | 3 | 3.33 | 3 | 4 | 3 | 3.33 |
| P80-FU2-SCAN-25m | 2 | 4 | 3 | 3.00 | 2 | 4 | 3 | 3.00 |
| P82-BASELINE-SCAN | 4 | 4 | 3 | 3.67 | 4 | 4 | 3 | 3.67 |
| P82-FU1-SCAN-23m | 4 | 4 | 4 | 4.00 | 4 | 4 | 4 | 4.00 |
| P82-FU2-SCAN-43m | 2 | 4 | 3 | 3.00 | 2 | 4 | 3 | 3.00 |
| P87-BASELINE-SCAN | 4 | 4 | 4 | 4.00 | 4 | 3 | 4 | 3.67 |
| P87-FU1-4M-SCAN | 4 | 4 | 4 | 4.00 | 4 | 4 | 4 | 4.00 |
| P87-FU2-14M-SCAN | 3 | 4 | 3 | 3.33 | 3 | 4 | 3 | 3.33 |
| P87-FU3-25M-SCAN | 3 | 3 | 3 | 3.00 | 3 | 4 | 3 | 3.33 |
| P87-FU4-40M-SCAN | 3 | 4 | 3 | 3.33 | 3 | 4 | 4 | 3.67 |
| P96-BASELINE-SCAN | 4 | 4 | 4 | 4.00 | 4 | 4 | 4 | 4.00 |
| P96-FU1-5M-SCAN | 3 | 4 | 3 | 3.33 | 3 | 4 | 3 | 3.33 |
| P96-FU2-12M-SCAN | 3 | 4 | 3 | 3.33 | 3 | 4 | 3 | 3.33 |
| P96-FU3-18M-SCAN | 3 | 4 | 4 | 3.67 | 3 | 4 | 4 | 3.67 |
| P96-FU4-24M-SCAN | 3 | 4 | 3 | 3.33 | 3 | 4 | 4 | 3.67 |
| P104-BASELINE-SCAN | 4 | 4 | 4 | 4.00 | 4 | 4 | 4 | 4.00 |
| P104-FU1-5M-SCAN | 4 | 4 | 4 | 4.00 | 4 | 4 | 4 | 4.00 |
| P104-FU2-15M-SCAN | 4 | 4 | 4 | 3.67 | 3 | 4 | 4 | 3.67 |
| P104-FU3-27M-SCAN | 2 | 4 | 2 | 2.67 | 4 | 4 | 3 | 3.67 |
| P104-FU4-41M-SCAN | 2 | 3 | 2 | 2.33 | 2 | 3 | 3 | 2.67 |
| P105-BASELINE-SCAN | 4 | 4 | 4 | 4.00 | 4 | 4 | 4 | 4.00 |
| P105-FU1-6M-SCAN | 4 | 4 | 4 | 4.00 | 4 | 4 | 4 | 4.00 |
| P105-FU2-12M-SCAN | 4 | 4 | 4 | 4.00 | 4 | 4 | 4 | 4.00 |
| P105-FU3-18M-SCAN | 4 | 4 | 4 | 4.00 | 4 | 4 | 4 | 4.00 |
| P106-BASELINE-SCAN | 3 | 4 | 3 | 3.33 | 3 | 4 | 2 | 3.00 |
| P106-FU1-6M-SCAN | 3 | 4 | 3 | 3.33 | 3 | 4 | 2 | 3.00 |
| P106-FU2-11M-SCAN | 4 | 4 | 4 | 4.00 | 4 | 4 | 4 | 4.00 |
| P107-BASELINE-SCAN | 4 | 4 | 4 | 4.00 | 4 | 4 | 4 | 4.00 |
| P107-FU1-SCAN-9m | 3 | 3 | 3 | 3.00 | 3 | 4 | 4 | 3.33 |
| P107-FU2-SCAN-26m | 3 | 3 | 3 | 3.00 | 3 | 3 | 4 | 3.33 |
| P109-BASELINE-SCAN | 4 | 4 | 4 | 4.00 | 4 | 4 | 4 | 4.00 |
| P109-FU1-SCAN-19m | 3 | 4 | 3 | 3.33 | 3 | 4 | 3 | 3.33 |
| P113-BASELINE-SCAN | 4 | 4 | 4 | 4.00 | 4 | 4 | 4 | 4.00 |
| P113-FU1-SCAN-4M | 4 | 4 | 4 | 4.00 | 4 | 4 | 4 | 4.00 |
| P113-FU2-SCAN-10M | 4 | 4 | 4 | 4.00 | 4 | 4 | 4 | 4.00 |
| P113-FU3-SCAN-17M | 4 | 4 | 4 | 4.00 | 4 | 4 | 4 | 4.00 |
| P136-BASELINE-SCAN | 3 | 4 | 4 | 3.67 | 3 | 4 | 4 | 3.67 |
| P136-FU1-9M-SCAN | 3 | 4 | 4 | 3.67 | 4 | 4 | 3 | 3.67 |
| P136-FU2-21M-SCAN | 2 | 4 | 4 | 3.33 | 4 | 4 | 4 | 4.00 |
| P138-BASELINE-SCAN | 4 | 4 | 4 | 4.00 | 3 | 4 | 4 | 3.67 |
| P138-FU1-2M-SCAN | 4 | 4 | 4 | 4.00 | 4 | 4 | 3 | 3.67 |
| P138-FU2-9M-SCAN | 5 | 4 | 4 | 4.33 | 4 | 4 | 4 | 4.00 |
| P144-BASELINE-SCAN | 4 | 4 | 4 | 4.00 | 4 | 4 | 4 | 4.00 |
| P144-FU1-13.9M-SCAN | 3 | 4 | 4 | 3.67 | 3 | 4 | 3 | 3.33 |
| P144-FU1-23M-SCAN | 3 | 4 | 4 | 3.67 | 4 | 4 | 4 | 4.00 |
| P144-FU1-3.2M-SCAN | 4 | 4 | 4 | 4.00 | 3 | 4 | 4 | 3.67 |

These relationships are visualized in Fig. 10 and Fig. 11.

## E.4. Clinical Viability

### E.4.1. Qualitative Evaluation (In-House Dataset)

The per-scan qualitative ratings of clinical viability on the in-house longitudinal dataset are summarized in Table 13.

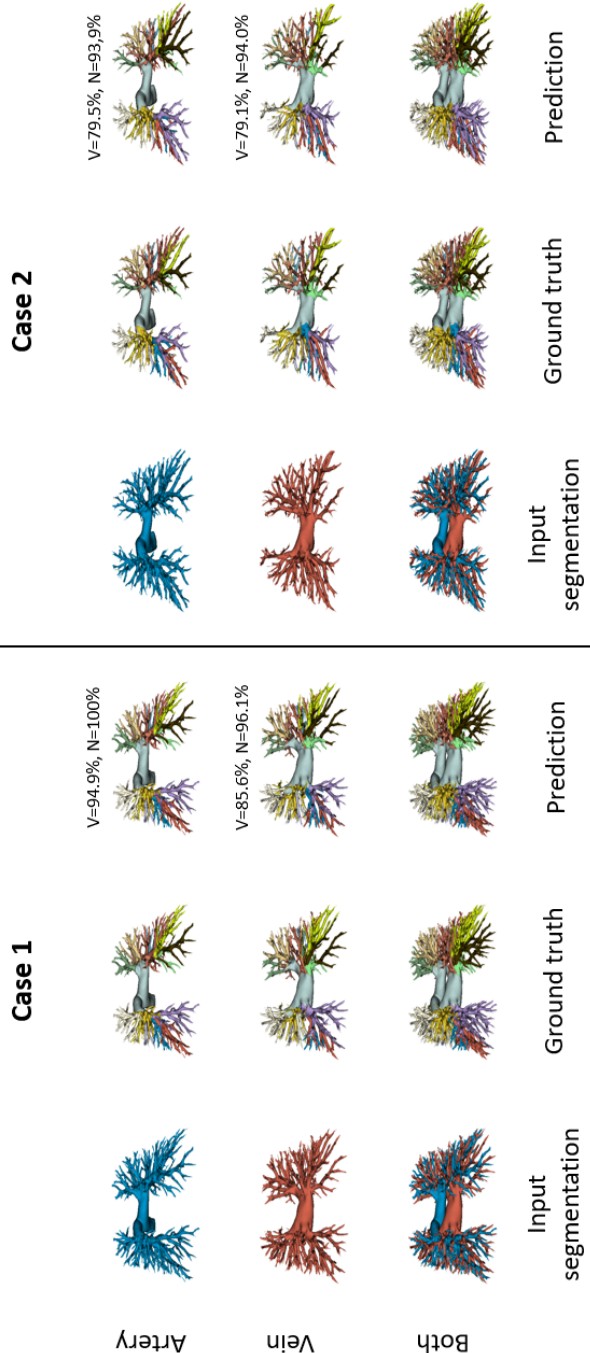

Figure 9: Representative anatomical labeling results for two PTL test cases. Case 1 demonstrates relatively high scores; Case 2 shows lower scores with mismatches in peripheral branches. All vessels are colored by anatomical class.

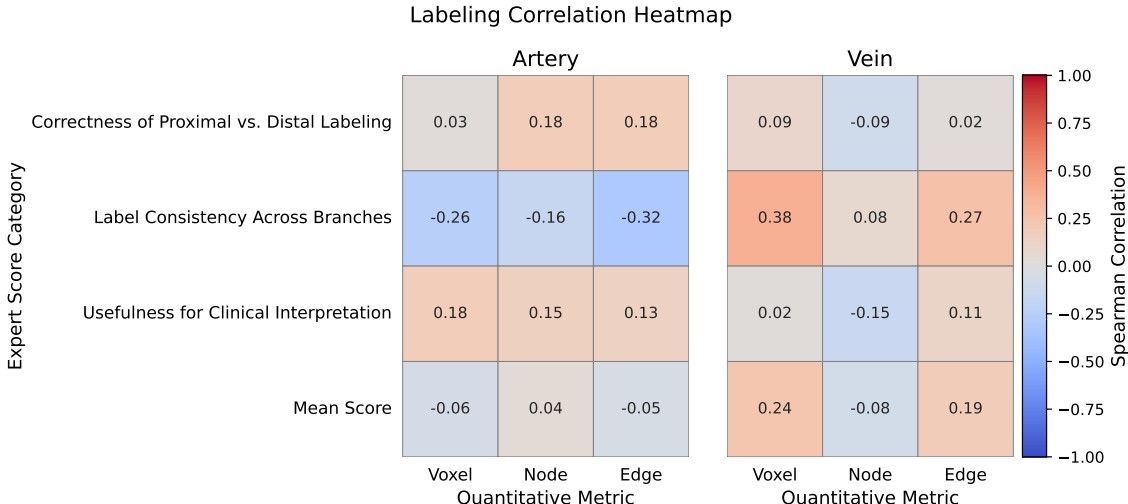

Figure 10: Spearman correlation between quantitative anatomical labeling metrics and expert-assigned scores across 21 PTL test scans, for arteries (left) and veins (right).

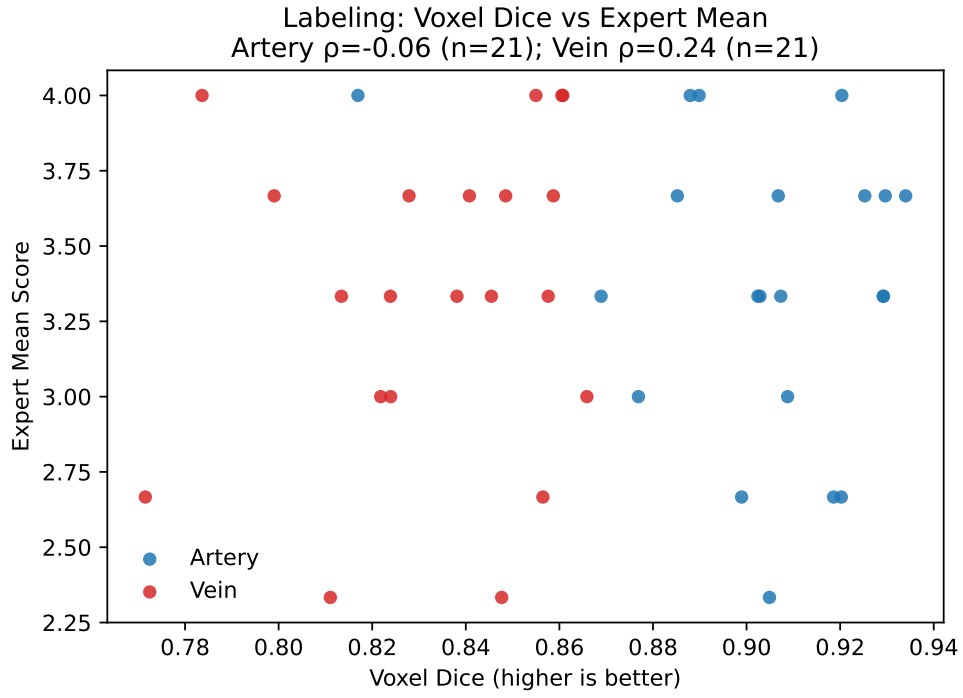

Figure 11: Case-level relationship between voxel-level labeling Dice and the expert mean score for arteries and veins.

