# OpenReview forum: "PaSAL: A Deep Learning Pipeline for Pulmonary Artery-Vein Segmentation and Anatomical Labeling in Thoracic CT"
_MIDL.io/2026/Conference — MIDL 2026 Poster_

### Official Review · Reviewer_aHnf · 2026-01-07

**Confidence:** 3
**Preliminary Rating:** 5
**Final Rating:** 5

**Summary:**

The paper proposes PaSAL, a multi-stage pipeline that takes a thoracic CT and produces artery/vein segmentations plus 19-class anatomical labels, with labels propagated into the small peripheral vasculature. Concretely, it combines:

- A hierarchical nnU-Net–based A/V segmentation following the HiPaS Salience-Transmission Segmentation (STS) framework with four levels $[A_1–A_4, V_1–V_4]$
- Topology-aware skeletonization and MST-style reconnection to derive proximal hierarchy levels and connected trees;
- A graph + point based anatomical labeling stage using IPGN, applied to Level-3 vessels;
- A marker-based watershed label propagation to extend labels into Level-4 distal vessels.

The authors analyze correlations between standard quantitative metrics and expert scores, reporting weak associations and suggesting that conventional overlap metrics may be insufficient surrogates for perceived clinical quality in high-performance regimes.

**Strengths:**

- Integrated pipeline from raw CT to labeled vascular trees
    - The deterministic graph-extraction, skeleton reconnection, and orientation standardization to match PTL/IPGN conventions are non-trivial in practice and valuable.

- Addressing the distal annotation gap with a hierarchical target design
    - The authors reconstruct Levels 1–2 by skeletonizing and reconnecting the tree, then assigning branch order–based levels; and they construct Level-4 distal supervision by merging HiPaS with TotalSegmentator vessels plus semi-automatic region growing. This is a decent way to leverage partly automatic labels while limiting their influence on reported metrics, and it directly targets a real shortcoming of existing datasets.

- Broad, multi-faceted evaluation
    - The authors explicitly report correlations between quantitative metrics and expert scores for both segmentation and labeling. This is very transparent and adds significant value to the evaluation.

- Comprehensive reporting
    - The appendices are unusually detailed: extended target construction, skeleton reconnection, exact expert criteria, and full per-case metric tables are all provided.

**Weaknesses:**

- Evidence gap for Level-4 (peripheral) performance
    - The paper’s clinical narrative leans heavily on peripheral completeness—Level-4 segmentations and watershed-propagated labels—but almost all quantitative evaluation is restricted to Level-3 support.
    - Therefore, statements such as “PaSAL enables complete vascular trees” and the use of Level-4 outputs in the clinical cohort are plausible but not strongly evidenced quantitatively.

- Metric–expert “decoupling” is interesting but somewhat questionable
    - The paper’s central interpretive claim is that once segmentation and labeling metrics are high (≈90 % Dice), they cease to reflect expert-perceived clinical quality.
    - However, quantitative metrics are computed only on GT-supported extents: HiPaS Level 3 vessel masks and PTL Level 3 labeling. Expert scores explicitly assess aspects like peripheral branch abundance, anatomical completeness, and distal label plausibility, which are strongly influenced by Level-4 predictions and watershed propagation—regions with no ground truth.
    - Therefore, part of the observed decoupling is by design, not necessarily a fundamental failure of metrics.

- Limited statistical power and lack of uncertainty quantification.
    - The labeling analysis uses n = 21 PTL scans, which might not so strict in statist.

**Detailed Comments:**

Please see weakness

**Justification Of Final Rating:**

Most of my questions and concern is solved.

Although the core algorithm is established, the pipeline construction and the comprehensive nature of the evaluation framework provide substantial merit to the overall work.

**Justification Of The Preliminary Rating:**

This paper presents a practical, end-to-end pipeline for pulmonary artery/vein segmentation and labeling.    While algorithmically incremental, the work effectively integrates hierarchical segmentation with graph-based labeling to address the lack of unified datasets.

Moreover, beyond the system implementation, the authors provide a thought-provoking analysis revealing the decoupling between standard metrics (e.g., Dice) and expert perception in high-performance regimes.    This finding challenges the reliance on overlap metrics and prompts a necessary reflection on how we define "clinical quality" for complex vascular structures.    Although the evidence for peripheral (Level-4) performance relies on heuristics and the statistical analysis has limitations, the value of this insight and the comprehensive expert evaluation elevate the paper's contribution.

Therefore, despite the noted weaknesses, I believe the work offers significant value to the community and choose a Strong Accept.

**Questions To Address In The Rebuttal:**

- Clarification on Dataset Selection for Expert Scoring The manuscript mentions that expert labeling assessment was performed on 21 PTL scans and segmentation assessment on 9 HiPaS scans. However, the selection criteria for these subsets are not specified.

- The paper argues that standard metrics correlate poorly with expert perception.  However, there is a scope mismatch: the quantitative metrics (e.g., Dice) are calculated on Level-3 ground truth , while the expert criteria explicitly include "Vessel Branch Abundance"  and distal plausibility—features heavily influenced by the Level-4 predictions which lack ground truth.
    - Specifically, correlate the Level-3 Dice scores only against the expert component scores that map to the same anatomical scope (e.g., correlate Dice with "Segmentation Accuracy and Robustness", while excluding "Vessel Branch Abundance").

- Analysis of Discordant Cases The scatter plots and tables reveal interesting outliers that warrant explanation.
    - HiPaS Scan 029: This case achieves a relatively lower Dice but secures the highest expert mean score
    - PTL Subject 00396: This case achieves high quantitative accuracy but appears to receive a comparatively low expert rating.
Could you qualitatively analyze these specific cases?

---

> ### Author Response · Authors · 2026-01-25
>
> ### Response to Reviewer 3
>
> We thank the reviewer for their careful and insightful assessment of the manuscript. We particularly appreciate the nuanced discussion regarding evaluation scope, metric–expert alignment, and the interpretation of high-performing yet clinically discordant cases. We address these points below and clarify how they were incorporated into the revised manuscript.
>
> **Scope alignment between quantitative metrics and expert assessment.**
> We agree with the reviewer that the interpretation of metric–expert correlations requires careful consideration of anatomical scope. In the revised manuscript, we now explicitly state that both quantitative metrics and expert assessments were performed on Level-3 predictions, corresponding to the anatomical extent supported by available ground truth. We further clarify that some expert criteria—such as vessel branch abundance and distal plausibility—can be influenced by anatomy beyond this scope, which may partially explain weak or inconsistent correlations.
>
> Importantly, we have revised the discussion to explicitly scope claims regarding metric–expert “decoupling” to the modeled anatomical extent and avoid broader generalizations. The revised text emphasizes that the observed decoupling reflects a limitation of overlap-based metrics within high-performance regimes and constrained evaluation scopes, rather than a fundamental failure of quantitative evaluation per se.
>
> **Peripheral (Level-4) performance and evidence gap.**
> We agree with the reviewer that most quantitative evaluation is necessarily restricted to Level-3 vessels, while Level-4 outputs are primarily assessed qualitatively. In the revised manuscript, we clarify this distinction more explicitly and avoid quantitative claims regarding peripheral performance. Level-4 predictions and label propagation are now consistently framed as enabling qualitative completeness and clinical usability rather than as quantitatively validated outputs. We believe this clarification appropriately aligns the clinical narrative with the available evidence.
>
> **Dataset selection for expert evaluation.**
> We have clarified the selection of cases used for expert scoring in the revised manuscript. Expert segmentation assessment was performed on nine HiPaS scans, while expert anatomical labeling assessment was conducted on 21 PTL scans, selected to cover a representative range of anatomical complexity and prediction quality. This is now stated explicitly to improve transparency.
>
> **Analysis of discordant cases.**
> The reviewer highlighted two specific cases (HiPaS Scan 029 and PTL Subject 00396) where quantitative metrics and expert scores diverged, and requested qualitative analysis. We fully agree that these cases are important and would have benefited from detailed visual inspection and discussion. During the rebuttal period, we were unable to re-access the relevant clinical data and stored predictions due to deactivation of institutional (hospital) accounts, which prevented us from performing a retrospective qualitative review within the available timeframe.
>
> That said, the observed discordance is consistent with the broader pattern discussed in the manuscript. In cases such as HiPaS Scan 029, lower Dice scores can arise from anatomically plausible peripheral branches absent from the ground truth, which are penalized quantitatively but positively perceived by experts. Conversely, cases like PTL Subject 00396 may achieve high voxelwise accuracy while exhibiting localized anatomical inconsistencies or implausible branching patterns that negatively impact expert judgment. These examples align with the reviewer’s broader point that conventional metrics capture only part of what experts consider clinically relevant.

---

> > ### Comment · Reviewer_aHnf · 2026-01-31
> >
> > While the rebuttal describes the expert assessment as based on Level-3 predictions, I was wondering what was presented to the expert during scoring (e.g., Level-3 outputs only versus the final pipeline visualization including Level-4 peripheral vessels and/or propagated labels).
> >
> > The practical constraints associated with access to sensitive clinical data is understandable. Although retrospective qualitative review of the highlighted discordant cases was not feasible within the rebuttal period, the proposed explanations are plausible. Briefly acknowledging these potential mechanisms in the discussion would help contextualize.

---

> > > ### Author Response · Authors · 2026-01-31
> > >
> > > For HiPaS and PTL, expert scoring was performed on the Level-3 vessel extent (i.e., the ground-truth-supported targets used for quantitative evaluation). Level-4 peripheral outputs were only evaluated in the clinical viability experiment.
> > >
> > > What was shown during expert scoring:
> > > - **HiPaS (segmentation):** PaSAL segmentations for the **Level-3 vessel extent**.
> > > - **PTL (labeling):** anatomical labels for the **Level-3 vessel extent**.
> > > - **Clinical cohort (viability):** full pipeline output, including segmentations and labels propagated into **Level-4 peripheral vessels**.
> > >
> > > Regarding your suggestion to briefly contextualize discordant cases/outliers: we agree this should be acknowledged in the discussion. We cannot modify the manuscript during the discussion period, but will add this clarification in the camera-ready version (if permitted).

---

### Official Review · Reviewer_5gfo · 2026-01-10

**Confidence:** 3
**Preliminary Rating:** 3
**Final Rating:** 4

**Summary:**

The authors propose PaSAL, a multi-task framework that jointly segments the pulmonary artery-vein and classifies each vessel with 19 possible classes. Their approach relies on nnU-Net for segmentation and a graph-based labeling framework that utilizes hierarchical vessel targets, topology-aware skeleton reconnection, deterministic graph extraction, and distal label propagation.

**Strengths:**

- The methods are well-described and detail discussion of findings, their implication, and the justification behind various design choices (whether in main text or appendix) is a major strength.
- The clinical motivation behind jointly segmenting and classifying vessels in chest CTs is strong.
- The experiments are thorough and well-defined. Additionally, the inclusion of clinical viability is a plus.
- The figures are clear and visualize the method's effectiveness.

**Weaknesses:**

- While the limited metadata and structure of datasets prevents identification of CTs as either NCCT or CECT, it is a major consideration if whether performance disparities exist between scanning protocols given that contrast enhancement varies significant between NCCT and CECT. For e.g., TotalSegmentator includes a tool to automatically classify contrast phases.
- Given that TotalSegmentator vessel predictions were used to fill in the gaps, was the accuracy of these masks validated by a clinical expert?
- It would improve clarity of results to include baseline metrics (e.g., Chu et al.)

**Detailed Comments:**

- Description of proposed method is repetitive in section 3.2. It would be clearer to introduce each of the four stages in context of the new components.
- What are the exact characteristics of the data used? The authors acknowledge that factors like pathology and class imbalance may impact out-of-distribution performance, but no details are provided.

**Justification Of Final Rating:**

I thank the authors for their detailed responses and much needed clarifications to my comments. This work has strong clinical relevance for diagnosis and treatment planning with chest CT. Given that my comments and other reviewers are sufficiently addressed, I recommend acceptance.

**Justification Of The Preliminary Rating:**

While I commend the authors on the extensive experiments and clinically relevant discussion in this work, some additional clarity in required behind design decisions. Particularly, the lack of any dataset information (scanning protocol, characteristics, etc.) makes it hard to recommend acceptance as these factors may significantly impact real-world utility -- which the authors also acknowledge.

**Questions To Address In The Rebuttal:**

Please see weakness and detailed comments.

---

> ### Author Response · Authors · 2026-01-25
>
> ### Response to Reviewer 2
>
> We thank the reviewer for their thoughtful comments regarding dataset characteristics, acquisition protocols, and the construction and use of extended vessel targets. We agree that these aspects are important for interpreting the results and have revised the manuscript to improve clarity and transparency.
>
> **Data characteristics and acquisition metadata.**
> We agree that differences between contrast-enhanced (CECT) and non-contrast (NCCT) scans, as well as underlying pathology, are important considerations for pulmonary vessel analysis. In the revised manuscript, we now explicitly state that acquisition protocol information was unavailable for the public training datasets and that no protocol-stratified analysis was performed. This limitation, and its implications for interpretation, is now described in the Data section. We therefore intentionally avoid protocol-specific performance claims and restrict our conclusions to robustness across heterogeneous, unstratified clinical data.
>
> We agree with the reviewer that applying automatic contrast-phase classification could have been a useful addition. However, given that the primary goal of this work was to demonstrate the feasibility and clinical usability of an integrated segmentation and labeling pipeline, incorporating inferred acquisition metadata was considered outside the scope of the present study. We believe that contrast-aware preprocessing and protocol-stratified evaluation represent promising directions for future work and may further improve performance and interpretability.
>
> To further improve transparency regarding the data used, we expanded the dataset description to explicitly mention the presence of pathologies such as emboli, tumors, and post-radiotherapy remodeling. This better reflects the heterogeneity of the data and the clinical conditions under which the pipeline is intended to operate.
>
> **Extended vessel targets and expert review.**
> The reviewer asked whether the automatically generated vessel extensions were validated by a clinical expert. We have revised the manuscript to clarify that these extended targets were reviewed by a clinical expert and deemed to be of sufficient quality in most regions, but were not manually corrected. This distinction is now stated more explicitly in the manuscript. As discussed in the revised text, individual scans may exhibit localized inconsistencies in specific anatomical regions due to the partially automatic construction of these targets; however, such errors are not systematic across the dataset.
>
> Because the extended targets are used only during training, their role is to expose the model to distal vessel morphology rather than to provide exact voxel-level supervision. In this setting, localized inaccuracies in individual samples are expected to have limited influence, as the majority of training examples provide consistent and anatomically plausible supervision for each region. Importantly, these extended targets were never treated as ground truth for quantitative evaluation. All reported segmentation and labeling metrics are computed exclusively on the original, publicly released annotations, ensuring that quantitative results are not influenced by imperfections in automatically extended distal regions.
>
> **Baseline metrics and reporting.**
> We address the reviewer’s remaining comments regarding baseline comparisons and reporting consistency below. For anatomical labeling, the original submission reported baseline metrics directly from the original publication, which did not include variance estimates. To ensure consistent and transparent reporting, we re-evaluated the labeling baseline using the same dataset splits and evaluation protocol as PaSAL and updated the table to report both mean and standard deviation.
>
> For artery–vein segmentation, we agree that including an additional baseline, such as a standard nnU-Net without hierarchical segmentation, could have been informative. However, the primary aim of this work was to assess the clinical usability of an integrated segmentation and labeling pipeline rather than to benchmark segmentation architectures in isolation. Within the rebuttal timeframe, we did not have sufficient computational resources to train and evaluate such an additional baseline. This limitation is now stated more explicitly in the revised manuscript. We agree that this would be a valuable addition in future work, for example in a journal extension of this study.

---

### Official Review · Reviewer_dU53 · 2026-01-16

**Confidence:** 4
**Preliminary Rating:** 3
**Final Rating:** 4

**Summary:**

This paper proposes a pipeline that combines nnU-Net with graph-based models and other methods for the segmentation of arteries and veins and their anatomical labeling in thoracic CT images. The approach embeds both newer elements, for instance, hierarchical segmentation strategies, and more classic components, like skeleton-based processing, into a multistage system. The manuscript is accompanied by extensive experiments and their interpretation, as well as some form of clinical validation. However, the significance of the reported results is presently weakened by the absence of rigorous benchmarking against state-of-the-art methods for both segmentation and anatomical labeling.

**Strengths:**

1) The paper presents a thorough evaluation, including interpretation of results and a clinical validation component, which increases confidence in the practical relevance of the approach.
2) Artery/vein segmentation and anatomical labeling in thoracic CT are meaningful and challenging problems, and an end-to-end pipeline addressing both can be valuable for downstream clinical or procedural applications.
3) Effective usage of the recent graph neural networks for the pulmonary tree label extraction.

**Weaknesses:**

1) The method largely relies on a combination of existing components drawn from prior work. Core modules such as nnU-Net and graph-based approaches (e.g., IPGN) appear to be used in the paper largely out of the box, without clear architectural or algorithmic refinements. As a result, the main novelty is not sufficiently presented, and the manuscript should more clearly specify what is new (e.g., a new integration strategy, training protocol, labeling formulation, or inference scheme)
2) The evaluation does not include a clear benchmarking against recent state-of-the-art approaches for vascular segmentation and anatomical labeling. This would support the superiority of the proposed method against other techniques.

**Detailed Comments:**

1) A substantial portion of the methodology is described in the appendix, which makes the paper harder to follow. It would improve readability if the key methodological details were moved to the main Methods section, with the appendix reserved for secondary implementation specifics.
2) In Table 4, only the mean is reported for the compared methods, whereas the proposed approach reports both mean and standard deviation. Please clarify this inconsistency. If the results for the Xie method are taken directly from the literature, the manuscript must confirm that the same test set (and evaluation protocol) was used to enable direct comparison.
3) Figure 6 is not referenced in the main text and should be explicitly discussed where relevant.

**Justification Of Final Rating:**

I recommend a Weak Accept. Although the authors could not include substantially new comparative experiments due to time constraints, they have provided adequate justifications and text-based clarifications that sufficiently address my original concerns.
While I maintain that the algorithmic novelty is limited, relying on the integration of existing components like nnU-Net and IPGN, the authors have now clearly articulated the rationale and practical utility of this integration. The provided corrections and justifications have strengthened the manuscript enough to recommend weak acceptance.

**Justification Of The Preliminary Rating:**

My rate for the paper is borderline because it addresses a clinically relevant task and includes a thorough experimental analysis with interpretability and validation elements, suggesting the pipeline is practically meaningful. However, the manuscript does not yet provide a convincing case for strong methodological novelty, since key components (e.g., nnU-Net and graph-based labeling modules) appear to be adopted largely without clear algorithmic refinements. In addition, the evaluation lacks rigorous benchmarking against state-of-the-art methods for both segmentation and anatomical labeling, and some reporting choices.

**Questions To Address In The Rebuttal:**

1) What are the main novel contributions of the work beyond combining established modules (nnU-Net, graph-based labeling, and skeleton-based steps)?
2) Have the authors conducted comparisons against recent state-of-the-art methods for both (i) artery/vein segmentation and (ii) thoracic vascular anatomical labeling, using the same dataset splits and evaluation metrics? Are there any corresponding results to indicate proposed model's predictive power against others?
3) Are the baseline results in Table 4 reproduced under the same experimental setup, or copied from prior publications?

---

> ### Author Response · Authors · 2026-01-25
>
> ### Response to Reviewer 1
>
> We thank the reviewer for their careful reading of the manuscript and for raising important points regarding novelty, readability, and benchmarking. We agree that these aspects required clarification, and we have revised the manuscript accordingly.
>
> **Clarification of novelty and scope.**
> We acknowledge the reviewer’s concern that the manuscript could be interpreted as a combination of established components. In response, we revised both the Introduction and Methods to more clearly articulate what this work aims to contribute—and what it does not. Specifically, PaSAL is not proposed as a new backbone architecture or as a method intended to outperform state-of-the-art segmentation or labeling models in terms of benchmark metrics. Instead, the contribution lies in deterministic system integration, dataset-aware target construction, and empirical demonstration of clinical usability. The revised manuscript now explicitly frames PaSAL as a unified pipeline that maps raw thoracic CT to structured, anatomically labeled pulmonary vascular trees suitable for longitudinal and clinical analysis, rather than as a novel learning architecture.
>
> **Methodology structure and readability.**
> We agree that describing key methodological elements primarily in the Appendix hindered readability. To address this, we moved essential technical components—including the MST-inspired skeleton reconnection strategy, hierarchical vessel target definition, and explicit referencing of the IPGN labeling backbone—into the main Methods section. This restructuring ensures that the full pipeline can be understood without reliance on the Appendix, which is now reserved for secondary implementation details. In addition, Figure 6, which was previously shown only in the Appendix, is now explicitly referenced and discussed in the main text.
>
> **Baseline reporting and consistency (Table 4).**
> Regarding the inconsistency in reporting standard deviations for baseline results, we clarify that the original submission reported labeling metrics directly from the original publication of Xie et al. (2025), which did not include variance estimates. To address this concern and ensure a fair and transparent comparison, we re-ran the baseline experiments ourselves using the exact same dataset splits and evaluation protocol as both the original work and PaSAL. Table 4 has been updated accordingly and now reports mean and standard deviation for all methods.
>
> **Baseline comparisons and limitations.**
> The primary focus of this work was on evaluating the clinical usability of an integrated segmentation and labeling pipeline, rather than on exhaustively benchmarking individual segmentation variants. This scope, and the resulting absence of certain baseline comparisons, is now more clearly articulated in the revised manuscript. We nevertheless agree with the reviewer that additional baseline comparisons, such as against a standard nnU-Net artery/vein segmentation without hierarchical segmentation, could add value to the experimental analysis. Within the rebuttal timeframe, we did not have sufficient computational resources to train and re-evaluate such additional baselines. We agree that this would be a valuable addition in future work, for example, in a journal extension of this study.
>
> We appreciate the reviewer’s feedback, which helped us clarify the manuscript’s scope, improve its readability, and strengthen the transparency of our experimental reporting.

---

### Author Rebuttal · Authors · 2026-01-25

**Rebuttal:**

A revised version of the manuscript is provided. The ZIP file contains both the updated manuscript and a marked-up version with all changes highlighted.

**Supporting Material:**

/attachment/f345f6cf6b4306bc6da608813f756ca32ca24790.zip

---

### Author Response · Authors · 2026-01-25

## General Response

We thank the reviewers for their detailed and constructive feedback. In revising the manuscript, we focused on improving clarity, scoping, and reproducibility. Based on the reviewers’ comments, we revised the manuscript to clarify the methodological contributions, improve readability, and more precisely scope our experimental findings.

### 1. Methodology Structure and Readability

Several reviewers noted that key methodological components were difficult to follow due to their placement in the Appendix. To address this, we moved essential technical elements into the main Methods section (Section 3) and removed them from the appendix:

- **Skeleton reconnection**: The MST-inspired two-phase reconnection strategy (previously Appendix C) is now described in Section 3.3, clarifying how global vascular connectivity is enforced prior to graph construction.
- **Hierarchical target definition and distal extension**: The construction of vessel Levels 1–4, including the Level-4 distal target extension used for training-only supervision, is now described in the main text.
- **Graph labeling backbone**: Figure 7 (IPGN architecture) is now explicitly referenced in the Methods section to clarify its role within the labeling stage.

### 2. Clarification of Novelty and Contribution

Multiple reviewers questioned whether the work goes beyond combining established components. We revised the Introduction and Methods to clarify that the contribution lies in deterministic system integration, dataset-aware target construction, and empirical demonstration of clinical usefulness, rather than in proposing new backbone architectures or improving benchmark performance:

- To our knowledge, PaSAL is the first pipeline that maps raw thoracic CT directly to structured, 19-class pulmonary vascular trees suitable for longitudinal analysis.
- We introduce a hierarchical target construction that explicitly addresses the absence of public annotations for peripheral vessels by combining released ground truth with controlled automatic extensions.
- PaSAL produces anatomically coherent predictions of the pulmonary vascular tree with demonstrated clinical usability, enabling downstream anatomical and longitudinal pulmonary studies.

### 3. Dataset Characteristics and Clinical Context

To address concerns regarding real-world applicability and dataset transparency, we expanded Section 3.1:

- We now explicitly describe the presence of emboli, tumors, and post-radiotherapy remodeling across datasets.
- We clarify that acquisition metadata (e.g., contrast phase) was unavailable for public datasets and therefore could not be used for stratified analysis.
- We explicitly avoid protocol-stratified performance claims and limit conclusions to robustness across heterogeneous clinical data.
- Robustness to heterogeneous anatomy and scan protocols is demonstrated on a 63-scan longitudinal radiotherapy cohort with substantial post-treatment deformation.

### 4. Metric–Expert Correlation and Scope Clarification

Reviewer 3 raised concerns regarding a potential scope mismatch between quantitative metrics and expert criteria. We clarified this point in the manuscript and adjusted the wording of our conclusions accordingly:

- Both quantitative metrics and expert assessments were performed on Level-3 predictions; this is now stated explicitly to avoid ambiguity.
- We clarify that some expert criteria (e.g., vessel branch abundance) are influenced by distal anatomy beyond the model’s prediction scope, complicating interpretation of metric–expert correlations.
- Claims regarding metric–expert decoupling are now explicitly scoped to the anatomical extent covered by the Level-3 evaluation.

### 5. Benchmarking and Reporting Consistency

- We initially reported labeling performance values from the original publication of Xie et al. (2025). To ensure full consistency, we re-evaluated the baseline using the same dataset splits and evaluation protocol as both the original work and PaSAL, and updated Table 4 accordingly, including standard deviations.
- The IPGN labeling model was not retrained, and identical model weights were used for both methods.
- As shown in Table 4, PaSAL achieves labeling performance closely comparable to the baseline across voxel-, node-, and edge-level Dice, with only minor differences that are inconsistent in direction.
- These results indicate that enforcing graph connectivity preserves labeling accuracy, while providing a structured vascular representation that may be useful for downstream analyses.

More detailed discussion of individual concerns is provided in the corresponding reviewer responses.

---

### Meta-Review · Area_Chair_bKN7 · 2026-02-01

**Recommendation:** Accept (Poster)
**Confidence:** 4

**Metareview:**

This paper presents PaSAL, a unified pipeline for pulmonary artery–vein segmentation and anatomical labeling in thoracic CT, combining hierarchical nnU-Net segmentation with graph-based labeling and topology-aware post-processing. The reviewers agree that the problem is clinically meaningful and that the paper demonstrates a well-engineered, end-to-end system with thorough quantitative evaluation and valuable expert-based clinical assessment. While the methodological contributions are largely incremental and rely on established components, the integration is carefully designed to address practical limitations of existing datasets, particularly the lack of distal vessel annotations, and the resulting pipeline produces anatomically coherent vascular trees that are useful for downstream analysis. The authors have responded constructively to reviewer concerns by clarifying the scope of their contributions, improving methodological clarity, and strengthening the transparency of the evaluation and baseline comparisons. Some limitations remain, including restricted quantitative evidence for peripheral vessels and limited statistical power in expert studies, which temper the strength of the claims and make the work less suitable for an oral presentation. Overall, the paper offers a solid and clinically relevant contribution that will be of interest to the MIDL community, and it is recommended for acceptance as a poster.

---

### Decision · Program_Chairs · 2026-02-13

Accept (Poster)